# Disentangling Geometry, Performance, and Training in Language Models

**Atharva Kulkarni** [1]  **Jacob Mitchell Springer** [2]  **Arjun Subramonian** [3]  **Swabha Swayamdipta** [1]

## Abstract

Geometric properties of Transformer weights, particularly the unembedding matrix, have been widely useful in language model interpretability research. Yet, their utility for estimating downstream performance remains unclear. In this work, we systematically investigate the relationship between model performance and the unembedding matrix geometry, particularly its effective rank. Our experiments, involving a suite of 108 OLMo-style language models trained under controlled variation, reveal several key findings. While the best-performing models often exhibit a high effective rank, this trend is not universal across tasks and training setups. Contrary to prior work, we find that low effective rank does not cause late-stage performance degradation in small models, but instead co-occurs with it; we find adversarial cases where low-rank models do not exhibit saturation. Moreover, we show that effective rank is strongly influenced by pre-training hyperparameters, such as batch size and weight decay, which in-turn affect the model's performance. Lastly, extending our analysis to other geometric metrics and final-layer representation, we find that these metrics are largely aligned, but none can reliably predict downstream performance. Overall, our findings suggest that the model's geometry, as captured by existing metrics, primarily reflects training choices rather than performance.

## 1. Introduction

As language models have grown in scale and complexity, interpretability efforts have increasingly turned to geometric characterizations of learned weights to understand model behavior (Nanda et al., 2023; Yunis et al., 2024). The study of the unembedding matrix – which maps hidden states to

[1]University of Southern California [2]Carnegie Mellon University [3]University of California, Los Angeles. Correspondence to: Atharva Kulkarni <atharva.kulkarni@usc.edu>.

*Proceedings of the 43$^{rd}$ International Conference on Machine Learning*, Seoul, South Korea. PMLR 306, 2026. Copyright 2026 by the author(s).

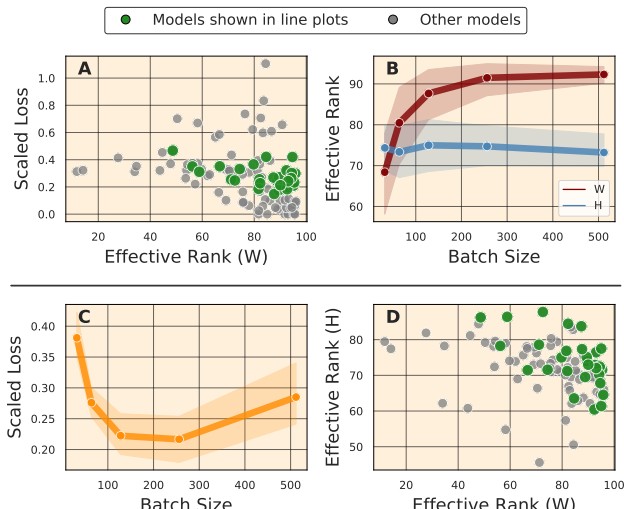

*Figure 1.* **(A)** The effective rank of a model's unembedding matrix, **W**, tends to correlate with its performance, but does not guarantee it. **(B)** The effective rank of the unembedding matrix is influenced by hyperparameter choices (e.g. batch size). **(C)** These hyperparameters in turn can affect model performance. **(D)** The effective rank of the last token's final-layer representation, **H** shows little correlation with that of the unembedding matrix and varies minimally across different hyperparameters (such as batch size) **(B)**. **Note:** The **green dots** are used to produce the line plots. The loss is normalized within each model size by its lowest observed value.

vocabulary logits – has attracted particular attention, as it can be analyzed consistently across architectures and provides a direct interface into how models produce predictions (Finlayson et al., 2024; Zhao et al., 2024; Zhou et al., 2024; Finlayson et al., 2026). Recent work has linked the geometric properties of this matrix, particularly its effective rank (Roy & Vetterli, 2007), to model performance; studies on the Pythia model family (Biderman et al., 2023) report that low effective rank (i.e., high concentration in the singular value spectrum) correlates with performance degradation (Godey et al., 2024b; Diehl Martinez et al., 2024b). These findings have led to the hypothesis that low rank in the unembedding matrix is harmful to language model performance.

However, we identify two critical gaps in this narrative. First, we argue that these findings may be limited in scope as they are based exclusively on the Pythia models. Second, prior work does not systematically vary training conditions to isolate *when* and *why* low effective rank emerges, making it unclear whether it has a *causal* effect on performance

or merely *co-occurs* with other confounding factors like skewed token distribution, optimization instability, or implicit regularization induced by hyperparameters. Motivated by these gaps, we ask a fundamental question: *RQ1) Does the effective rank of the unembedding matrix reliably predict model performance across tasks and training configurations?* Towards answering this, we pre-train a suite of 108 OLMo-style language models (Groeneveld et al., 2024) varying their size, pretraining token budget, and training hyperparameters. We then analyze the effective rank of each model's unembedding matrix alongside its performance on pretraining, out-of-distribution generalization, fine-tuning, knowledge retention, and post-training quantization. We find that **effective rank tends to correlate with performance, but does not guarantee it** (§3). Consistent with prior findings (Diehl Martinez et al., 2024b), we find that higher effective rank generally corresponds to better downstream loss. However, this relationship is not universal; We find multiple models with high effective rank that yield poor results, while some low-rank ones maintain good performance. We also revisit the *saturation* problem of small language models, where extended pretraining leads to late-stage performance degradation, especially in the Pythia models (Biderman et al., 2023; van der Wal et al., 2025). While prior work links this behavior to low effective rank (Godey et al., 2024b), our OLMo models maintain stable performance even under extreme over-training, including cases of severely low effective rank. This indicates that **low effective rank is neither necessary nor sufficient to explain saturation.**

A natural follow-up question arises: *RQ2) Why may the effective rank not correlate well with performance?* We find that **effective rank is strongly shaped by training hyperparameters – particularly batch size and weight decay – which can potentially confound the observed association between model performance and effective rank** (§4). For instance, we find that larger batch sizes consistently result in high effective rank, albeit without guaranteeing performance gains. Conversely, lower weight decay substantially reduces effective rank, while its effect on performance remains inconsistent.

Lastly, prior work has also analyzed the final-token's last-layer representations (Godey et al., 2024a; Machina & Mercer, 2024; Li et al., 2025), and used other geometric metrics like cosine similarity (Gao et al., 2019; Ethayarajh, 2019) and isotropy (Arora et al., 2016), without clearly distinguishing between these choices or establishing their relationships. This motivates our final research question: *RQ3) Beyond effective rank, how do other geometric metrics relate to performance, and do these relationships extend to learned representations?* Our findings suggest that other **alternative geometric metrics offer limited additional insight beyond those provided by effective rank in the settings**

we consider (§5). In particular, the effective rank of the unembedding matrix maintains a monotonic but non-linear correlation with cosine similarity, thus providing no additional discernment. Isotropy remains largely uncorrelated with other metrics and varies over a narrow range irrespective of model performance, hence limiting its utility as a diagnostic tool. Likewise, last-layer representations are relatively stable across training configurations and do not account for observed performance differences.

Taken together, our results raise questions about the view of unembedding geometry as a proxy for model performance. Our empirical findings suggest that effective rank and related metrics may primarily reflect optimization dynamics shaped by training hyperparameters. Accordingly, geometric measures are best used as diagnostic tools for understanding training behavior, rather than for model selection or performance prediction, and should be compared only across models trained under matched training settings and alongside task-based evaluations.

## 2. Background: Effective Rank and Model Geometry

Our main study focuses on analyzing the unembedding matrix of Transformers, henceforth referred by $\mathbf{W} \in \mathbb{R}^{v \times d}$, where $v$ is the vocabulary size and $d$ the model dimension. Since $\mathbf{W}$ maps hidden representations to token logits, its spectral structure characterizes how information is distributed across predictive directions in the hidden space. As in previous work (Godey et al., 2024b; Diehl Martinez et al., 2024b), we quantify this structure using the effective rank. Following Roy & Vetterli (2007), we define it as:

$$\mathcal{R}(\mathbf{W}) = \exp\left(-\sum_{k=1}^{\min(v,d)} p_k \log p_k\right) \quad (1)$$

where $p_k = \frac{\sigma_k}{\|\sigma\|_1} + \epsilon$ and $\sigma_i$ are the singular values of $\mathbf{W}$. We normalize $\mathcal{R}(\mathbf{W})$ by $\min(v, d)$ so that $\mathcal{R}(\mathbf{W}) \in [0, 1]$.

Beyond effective rank, prior work (Godey et al., 2024a; Machina & Mercer, 2024) has also examined the following two geometric metrics:

**Isotropy.** We use the partition-function-based metric defined by Arora et al. (2016):

$$\mathcal{I}(\mathbf{W}) = \frac{\min_{c \in \mathbf{U}} \mathbf{Z}(c)}{\max_{c \in \mathbf{U}} \mathbf{Z}(c)} \quad (2)$$

Here $\mathbf{U}$ is the eigenvector of $\mathbf{W}^\top \mathbf{W}$, $c$ is a unit vector, and $\mathbf{Z}(c) = \sum_i^v \exp(\langle c, \mathbf{W}_i \rangle)$. $\mathcal{I}(\mathbf{W}) \in [0, 1]$, with $\mathcal{I}(\mathbf{W}) = 1$ suggesting high isotropy. It quantifies how uniformly vectors are distributed across principal directions.

**Angular Variability.** Based on Ethayarajh (2019)'s formu-

lation we define it as:

$$\mathcal{A}(\mathbf{W}) = \frac{1}{|\mathbf{W}|^2 - |\mathbf{W}|} \sum_{w_i, w_j \in \mathbf{W}, i \neq j} \frac{w_i^T w_j}{\|w_i\|_2 \cdot \|w_j\|_2} \quad (3)$$

where $\mathcal{A}(\mathbf{W}) \in [-1, 1]$, with $\mathcal{A}(\mathbf{W}) = 1$ indicating low angular variability and high semantic similarity. We revisit these metrics, and their relationship to effective rank in §5.

**Impact of Model Geometry on Performance.** In general, low effective rank and high anisotropy have been considered detrimental, with recent works linking them to the performance degradation phenomenon in smaller Pythia models (Machina & Mercer, 2024; Diehl Martinez et al., 2024b; Godey et al., 2024b). High cosine similarity among the embedding vectors has long been linked to the *representation degeneration problem*, wherein model representations tend to become increasingly similar, collapsing into a narrow cone (Gao et al., 2019; Zhang et al., 2020; Biś et al., 2021). Diehl Martinez et al. (2024a)'s work indicates that representation anisotropy is positively correlated with the frequency bias issue. On the other hand, Li et al. (2025) find an opposite trend, with low effective rank leading to high n-gram memorization, and high effective rank enhancing generalization and learning of long-range dependencies. Moreover, Ding et al. (2022) find that reinstating isotropy in models often yields statistically insignificant gains over their anisotropic counterparts. Similarly, Ait-Saada & Nadif (2023) and Mickus et al. (2024) illustrate how anisotropy is not detrimental for clustering tasks.

Taken together, the evidence paints a fragmented and at times contradictory picture. Moreover, the role of these metrics in important tasks such as pre-training, generalization, forgetting, and quantization remains largely unexplored. Our work aims to bring coherence to this landscape by systematically analyzing Transformer geometry across definitions, tracing its causes, and clarifying if and when it matters for downstream performance.

**Causes of Geometric Degradation.** A prominent hypothesis links rank collapse and anisotropy to the long-tailed distribution of tokens in the data (Gong et al., 2018; Gao et al., 2019). Under this view, the frequent tokens dominate gradient updates, overpowering the rare tokens, causing push-pull dynamics during optimization, and ultimately leading to rank collapse (Biś et al., 2021; Yu et al., 2022; Mircea et al., 2024). Other work argues that anisotropy is not merely a byproduct of data distributions, but is instead inherent to the Transformer architecture (Godey et al., 2024a). Moreover, a suite of theoretical work hypothesizes that self-attention is the cause of rank collapse in the intermediate token representations (Dong et al., 2021; Noci et al., 2022; Zhai et al., 2023; Nait Saada et al., 2025). On the other hand, Cai et al. (2021) show that transformer em-

beddings can be isotropic, albeit only within small clusters or low-dimensional manifolds. Likewise, Timkey & van Schijndel (2021) argue that certain *rogue dimensions* dominate anisotropy calculations. More recently, Stollenwerk & Stollenwerk (2025) claim that second-moment estimates of the AdamW optimizer are a major contributor to embedding anisotropy. Despite these insights, prior work does not systematically study how these metrics vary with respect to different training hyperparameters.

Given this background, our investigation proceeds in three stages. First, we assess whether effective rank reliably predicts performance across diverse evaluation settings (§3). Second, we investigate which training factors (particularly hyperparameters) may drive variations in effective rank (§4) Finally, we examine how isotropy and angular variability relate to effective rank (§5), and whether these relationships extend from the unembedding matrix to learned representations (Appendix A.3).

## 3. Does Effective Rank Reliably Predict Model Performance?

We answer our *RQ1* here by examining the relationship between the effective rank of the unembedding matrix, $\mathcal{R}(\mathbf{W})$, and task performance. We use scatter plots to visually assess these relationships. Appendix A.3 contains detailed regression analysis results that support our observations in this section. Taken together, our empirical findings suggest that effective rank is predictive but not deterministic of downstream performance.

### 3.1. Experimental Setup

We train a suite of OLMo-style models (Groeneveld et al., 2024) on the Pile dataset (Gao et al., 2020) following Magnusson et al. (2025)'s configuration. Our experiments span model sizes of $N \in \{4, 8, 16, 28, 40, 75\}$ million non-embedding parameters trained on $D \in \{8, 32, 64, 128\}$ billion tokens. To study the effect of training hyperparameters, we perform systematic ablations. We vary the batch size over $\{32, 64, 128, 256, 512\}$, the weight decay over $\{0.1, 0.5, 0.01\}$, and the learning rate scaled to $\{0.1, 10, 100\}$ of the model-specific learning rate (Porian et al., 2024). Additionally, we vary the learning rate decay to 0, 1%, and 10% of the peak learning rate. This results in a suite of 108 pre-trained models for our study. More fine-grained model and training details are provided in Appendix A.1.

### 3.2. Results on Pretraining Performance

**Higher effective rank generally associates with lower in-distribution loss, but with substantial variance.** We evaluate our OLMo models on the Pile-10k corpus (Nanda, 2022) to measure in-distribution loss. Figure 2 shows the

relationship between model performance and the effective rank of the unembedding matrix $\mathbf{W}$. Across model sizes, we observe a consistent trend: lower loss is generally associated with higher effective rank. However, this relationship is not strictly monotonic; we observe numerous instances in which models with nearly identical losses exhibit markedly different rank profiles. For example, OLMo-28M models with loss near 5 span $\mathcal{R}(\mathbf{W})$ from 40 to 80, and OLMo-4M models with loss around 6.5 range from 10 to 80. Moreover, the worst-performing models are not the most collapsed; they correspond to models trained with high learning rates ($\approx$1e-2) and strong annealing (10%). Interestingly, across model sizes, we also observe many underperforming models with high effective rank. Closer inspection reveals that these are the models trained on smaller token budgets (8B, 32B) with large batch sizes (256, 512). We elaborate on the impact of hyperparameter choices on $\mathcal{R}(\mathbf{W})$ in Section 4.

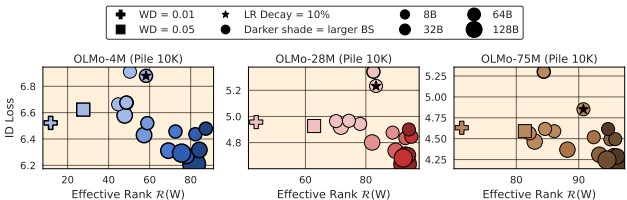

*Figure 2.* **In-distribution evaluation:** Variation in effective rank does not translate to equivalent change in in-domain loss.

**Out-of-distribution performance tracks in-distribution loss, not effective rank.** We use the Paloma benchmark (Magnusson et al., 2024) to assess OOD performance. Figure 3 shows results on the Dolma-100 programming-languages subset. Overall, OOD loss closely mirrors ID loss, while variation in effective rank $\mathcal{R}(\mathbf{W})$ is not consistently correlated with OOD losses. We also observe an *agreement-on-the-line* phenomenon (Miller et al., 2021; Baek et al., 2022), where models exhibit a near-linear relationship between ID and OOD loss. This suggests that ID loss is a more reliable predictor of OOD behavior than effective rank.

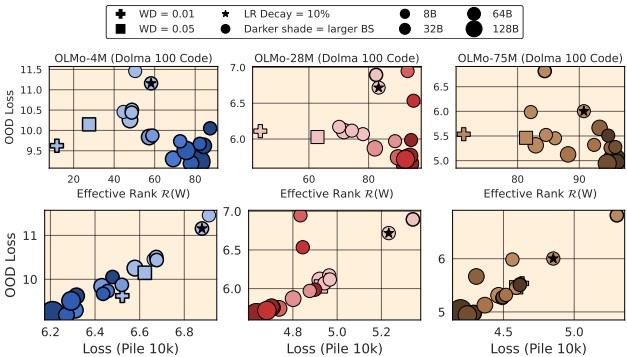

*Figure 3.* **Out-of-Distribution Evaluation:** Performance in OOD setup remains largely unchanged across different effective ranks $\mathcal{R}(\mathbf{W})$. Alternatively, models with stronger in-domain loss consistently transfer well, making it a better predictor of OOD behavior.

**Quantization robustness shows scale-dependent sensitivity to effective rank.** We use the GPTQ framework (Frantar et al., 2023) to post-train quantize our models and assess their ID performance. Figure 4 shows that OLMo-4M exhibits a near-monotonic relationship between effective rank $\mathcal{R}(\mathbf{W})$ and post-quantization loss. With a few outliers, OLMo-28M models also show increased loss degradation as rank decreases, whereas the results for OLMo-75M remain comparatively inconsistent at high $\mathcal{R}(\mathbf{W})$. In contrast, the latter models show a strong *agreement-on-the-line* with their un-quantized counterparts. However, this alignment breaks for the smaller OLMo-4M models, which show substantially more rank collapse. Notably, its 128B-token variant performs worse than models trained on smaller token budgets. This aligns with prior findings that overtraining small models degrades post-training performance (Kumar et al., 2025). We further observe that OLMo-4M models trained with low weight decay achieve good pre-quantization performance despite extremely low effective rank, but suffer severe post-quantization degradation. This suggests that quantization-induced degradation is more severe in extreme cases of collapse in $\mathbf{W}$. We find that these results also hold true for OOD evaluation (refer to Appendix A.3 Figure 23).

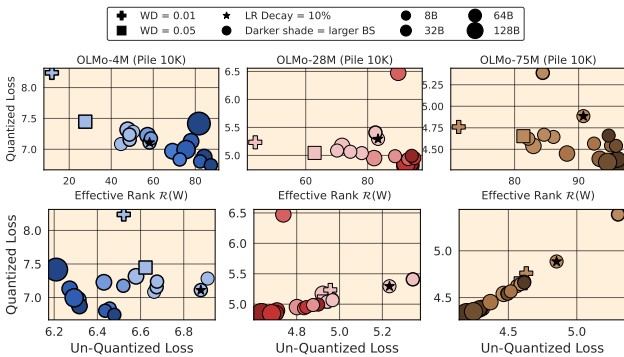

*Figure 4.* **Post-training quantization evaluation:** Highly collapsed models are heavily impacted, while high-rank models remain largely intact but not entirely consistent.

**Saturation is not caused by low effective rank alone.** Smaller Pythia models ($\leq$ 160M parameters) are known to exhibit performance degradation toward the end of pre-training (Biderman et al., 2023; van der Wal et al., 2025), a phenomenon commonly referred to as *saturation*. Prior work causally links it to the low rank of the unembedding matrix (Godey et al., 2024a; Machina & Mercer, 2024; Razzhigaev et al., 2024). However, existing studies focus almost exclusively on the Pythia family, leaving open two key questions: 1) Is saturation a general phenomenon across small language models? 2) Is a low effective rank the underlying cause? To disentangle these factors, we pre-train an OLMo-14M model that exactly matches the Pythia-14M configuration. The sole architectural difference is the use of sequential attention instead of its parallel variant (Shoeybi

et al., 2020) In Figure 5, we clearly see that Pythia-14M exhibits in-distribution loss degradation during the tail end of training. This also happens to coincide with a sharp drop in the effective rank $\mathcal{R}(\mathbf{W})$. In contrast, OLMo-14M trained under the same conditions shows neither loss degradation nor effective rank collapse. Interestingly, when we train OLMo-14M with a smaller token budget (8B tokens) and a reduced batch size of 32, we observe substantially lower effective rank (rightmost subplot of Figure 5), yet still no degradation in loss. These results point to three insights. First, saturation is not universal: not all small language models degrade under extreme over-training. Second, a low effective rank alone is insufficient to explain saturation, as a model can collapse substantially without exhibiting loss degradation. Third, the dynamics of effective rank may matter: degradation in Pythia-14M coincides with an abrupt drop in $\mathcal{R}(\mathbf{W})$, whereas collapse in OLMo-14M-8B is markedly more gradual. We conjecture that Pythia's parallel attention, which can underperform sequential attention (Chowdhery et al., 2023), may contribute to this behaviour. Moreover, we observed that the spectral norm of Pythia-14M was substantially higher than that of our OLMo-14M variants, suggesting that spectral capping methods (Takase et al., 2025; Newhouse et al., 2025) may mitigate this issue. We leave a deeper investigation into the root causes of saturation to future work.

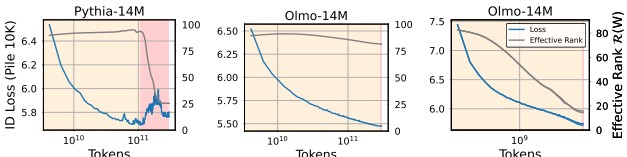

*Figure 5.* **Saturation evaluation:** Not all small language models experience saturation. Low effective rank alone does not cause it. Instead, saturation may result from other training or architectural factors that can also induce rank collapse.

> **Takeaways:** **1)** Models with ↑ $\mathcal{R}(\mathbf{W})$ have better ID and OOD performance, albeit it is not predictive across all training setups. **2)** ID loss is a better indicator of OOD behaviour than effective rank $\mathcal{R}(\mathbf{W})$. **3)** Models with ↓ $\mathcal{R}(\mathbf{W})$ consistently suffer more post-quantization. **4)** ↓ $\mathcal{R}(\mathbf{W})$ is not exclusively responsible for saturation in small language models.

### 3.3. Results on Fine-Tuning Performance

**Effective rank does not reliably predict fine-tuning performance.** We fine-tune OLMo models on the StarCoder-Python dataset (Li et al., 2023) to assess whether differences in the unembedding matrix's effective rank affect the learnability of new knowledge. Similar to previous cases, from Figure 6, we find that the best-performing models tend to exhibit high $\mathcal{R}(\mathbf{W})$. However, we observe many counterex-

amples in which models with comparable or even higher effective rank incur substantially worse fine-tuning loss. Moreover, unlike the pre-training evaluation results, we do not observe a strong *agreement-on-the-line* between fine-tuning and pre-training losses. We observe consistent trends across additional fine-tuning datasets (see Appendix A.3).

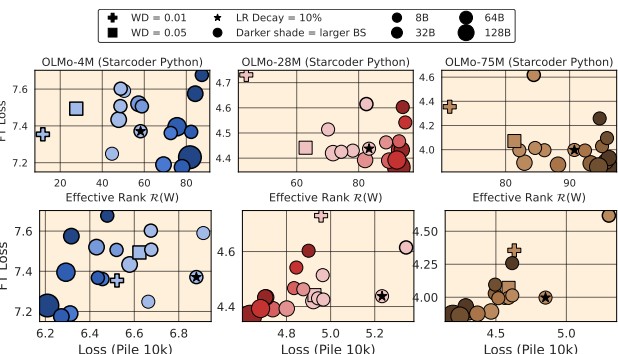

*Figure 6.* **Fine-tuning evaluation:** Neither the effective rank $\mathcal{R}(\mathbf{W})$ nor pre-training loss reliably predicts fine-tuning performance.

**Catastrophic forgetting is largely independent of effective rank.** Smaller models are particularly susceptible to catastrophic forgetting (Ramasesh et al., 2022), where fine-tuning degrades performance on the pre-training distribution. We investigate whether models trained with varied effective rank at initialization, affect catastrophic forgetting. We analyze this by evaluating the fine-tuned models on Pile-10k and measuring both absolute loss and change relative to base models (Δ loss). Interestingly, from Figure 7 we see that the fine-tuned models yield very similar in-distribution loss, largely independent of the base model's $\mathcal{R}(\mathbf{W})$ or its pre-training loss. Moreover, the Δ loss appears to correlate with the base model's ID loss, suggesting that fine-tuning performance is almost initialization-invariant. Surprisingly, across all models, we see that lower-weight-decay models, which exhibit the most collapse, tend to perform best even after fine-tuning. We hypothesize that the low-rank nature preserves the most representative components of the ID data, and hence forgets less. Taken together, these results suggest that fine-tuning induces a similar degree of learning (and forgetting) across models.

> **Takeaways:** Both Catastrophic forgetting and fine-tuning performance is not correlated with effective rank $\mathcal{R}(\mathbf{W})$ as well as pre-training performance.

## 4. Why Does Effective Rank Not Correlate Well with Performance?

In this section, we examine the factors that may drive variations in the effective rank $\mathcal{R}(\mathbf{W})$ in order to address *RQ2*. We find that $\mathcal{R}(\mathbf{W})$ is highly dependent on certain training

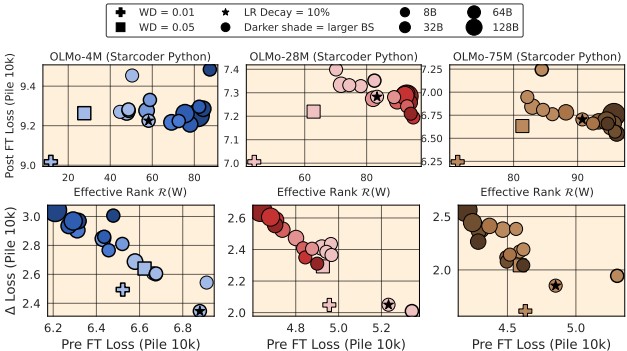

*Figure 7.* **Catastrophic forgetting evaluation:** While models with higher effective rank $\mathcal{R}(\mathbf{W})$ exhibit slightly larger relative forgetting ($\Delta$ loss), all fine-tuned models converge to similar absolute loss on the pre-training distribution. This suggests that catastrophic forgetting is largely independent of unembedding geometry.

hyperparameters, and these choices in turn influence model performance. To isolate these effects, we systematically ablate key hyperparameters (see §3.1) while keeping all others fixed, and evaluate both $\mathcal{R}(\mathbf{W})$ and in-distribution loss [1] .

## 4.1. Training Hyperparameters Influence Effective Rank

**Increasing batch size increases effective rank, without guaranteeing performance improvement.** Figure 8 shows that batch size has a pronounced effect on the geometry of $\mathbf{W}$. Across model sizes, small-batch training consistently leads to stronger collapse ($\downarrow \mathcal{R}(\mathbf{W})$), while larger batch sizes better preserve the effective rank ($\uparrow \mathcal{R}(\mathbf{W})$). The collapse in small batch size training regime is plausible, as gradients are dominated by frequent tokens, whose repeated and aligned updates push the rows of $\mathbf{W}$ toward similar directions, while gradients from rare tokens are sparse and inconsistent (Biś et al., 2021; Yu et al., 2022; Mircea et al., 2024). The effect is further amplified on datasets such as the Pile, which exhibits a heavy-tail of low-frequency tokens (Elazar et al., 2024). We also observe that the effective rank varies only marginally with batch size in larger models. We hypothesize that this is because larger models inherently exhibit a richer singular value spectrum due to their higher dimensionality $d$. Consequently, achieving a degree of rank collapse comparable to that of a smaller model requires suppressing substantially more dimensions. For example, a 50% collapse corresponds to eliminating 32 dimensions when $d = 64$, but 384 dimensions when $d = 768$. This likely makes larger models less sensitive to hyperparameter-induced variations in effective rank. In terms of performance, we observe a *U-shaped* relationship between batch size and in-distribution loss, indicating the presence of a '*critical batch size*', beyond which further

increases do not reliably improve performance (McCandlish et al., 2018; Zhang et al., 2019). This also explains why, in §3.2, models with higher effective rank sometimes exhibit higher loss. Since all other hyperparameters are held fixed, and large-batch training is known to be particularly sensitive to hyperparameter tuning (Marek et al., 2025), we conjecture that the slight performance degradation at large batch sizes is likely an optimization artifact that could be mitigated by learning-rate retuning. Overall, our results indicate that in-distribution performance is primarily driven by the choice of pre-training batch size, rather than by the effective rank $\mathcal{R}(\mathbf{W})$.

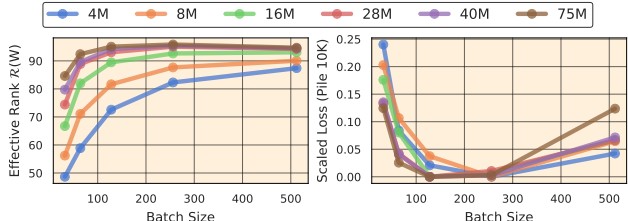

*Figure 8.* **Effect of batch size on $\mathcal{R}(\mathbf{W})$ and performance:** Larger batch sizes preserve higher effective rank in the unembedding matrix. However, performance exhibits a U-shaped dependence on batch size, with extremely large batches not always improving loss, explaining why higher $\mathcal{R}(\mathbf{W})$ does not necessarily translate to better performance.

**High regularization prompts high effective rank, but varied performance** Figure 9 suggests that weight decay also has a systematic effect on the unembedding matrix's geometry. We find that stronger regularization consistently preserves high effective rank $\uparrow \mathcal{R}(\mathbf{W})$. This can be attributed to weight decay's role in controlling the effective learning rate (D' Angelo et al., 2024) and promoting a rotational equilibrium that counteracts highly imbalanced gradient updates (Kosson et al., 2024). In contrast, the impact of weight decay on performance is not uniform across model sizes. Smaller models (4M, 8M) perform best under weak regularization, likely because stronger weight decay restricts their already limited capacity. As model size increases, representational capacity grows, and we observe improved performance with moderately higher regularization. Consequently, models of different sizes can exhibit both strong and weak performance based on their regularization strength while maintaining high effective rank. This further illustrates that a high $\mathcal{R}(\mathbf{W})$ is neither sufficient nor universally desirable for good performance, reinforcing the disconnect between model geometry and loss.

**Learning Rate affects effective rank and performance inconsistently.** As mentioned in §3.1, we choose the optimal learning rate for each model based on the scaling laws of Porian et al. (2024). Thus, each model has a different learning rate. To study learning rate sensitivity, we ablate these values by scaling them by factors of 0.1, 10, 100, yielding

---

[1]For a fair comparison across model sizes, we normalize the loss per model size, hereafter termed the scaled loss.

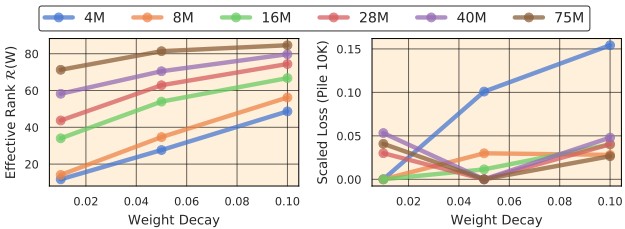

*Figure 9.* **Effect of weight decay on $\mathcal{R}(\mathbf{W})$ and performance:** Stronger regularization consistently preserves higher effective rank. Its impact on loss depends on model size, yielding both good and poor performance at similar $\mathcal{R}(\mathbf{W})$.

learning rates in the range $[10^{-4}, 10^{-2}]$. For comparability, we normalize learning rates by the smallest value per model size and examine their relationship with effective rank $\mathcal{R}(\mathbf{W})$ and in-distribution loss. From Figure 10, we can see that the effect of learning rate is nuanced. While $\mathcal{R}(\mathbf{W})$ remains largely stable across learning rates, smaller models tend to exhibit slight rank reduction, whereas larger models show mild rank expansion. In contrast, performance is strongly learning-rate dependent and diverges by scale. Smaller models benefit from higher learning rates, whereas larger models degrade at higher ones. Larger models show the opposite trend: they perform worse at high learning rates and improve as the learning rate decreases. These behaviors align with prior observations and motivate the use of $\mu$P scaling (Yang et al., 2021) to better align optimization dynamics across model sizes. Importantly, the learning rates prescribed by Porian et al. (2024) consistently yield the best performance across all models. Nevertheless, across learning rates, we observe models with well-preserved unembedding geometry but varying loss, once again highlighting the limitations of using $\mathcal{R}(\mathbf{W})$ alone as a predictor of performance.

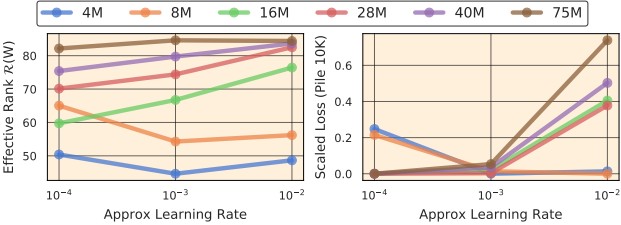

*Figure 10.* **Effect of learning rate on $\mathcal{R}(\mathbf{W})$ and performance:** Smaller models perform better with higher learning rates, whereas larger models benefit from lower learning rates. However, variations in $\mathcal{R}(\mathbf{W})$ remain minimal.

**High learning rate annealing leads to high effective rank and low performance.** Figure 11 shows that learning rate annealing also affects the geometry of the unembedding matrix. Aggressive schedules that decay the learning rate to 0–1% of its initial value have little effect on the effective rank $\mathcal{R}(\mathbf{W})$. In contrast, milder decay to 10% consistently preserves higher effective rank. In contrast, in-distribution performance degrades monotonically as the final learning

rate increases, indicating that maintaining a relatively high learning rate toward the end of training is suboptimal. As a result, this setting exhibits a clear inverse trend: configurations with higher $\mathcal{R}(\mathbf{W})$ (induced by less aggressive annealing) tend to incur higher loss. This further reinforces our central observation that a high effective rank does not necessarily translate to better performance.

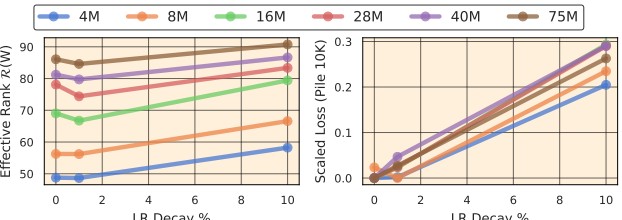

*Figure 11.* **Effect of LR decay on $\mathcal{R}(\mathbf{W})$ and performance:** Annealing the learning rate toward 0 improves performance, while changes in $\mathcal{R}(\mathbf{W})$ remain minimal.

**Training on more tokens preserves a high effective rank and high performance.** To isolate the effect of data scale, we train OLMo-style models on token budgets ranging from 8B to 128B tokens. To prevent extremely long schedules, we make sure that all of them are trained for the same steps, hence, with varying batch sizes. Prior work suggests that pre-training on tokens way beyond the *Chinchilla optimal* (Hoffmann et al., 2022) can continue to improve performance (Sardana et al., 2024; Gadre et al., 2025). Our results from Figure 12 corroborate these findings, with models of all sizes showing better performance when trained on larger token budgets. We also see that models exhibit a high effective rank $\mathcal{R}(\mathbf{W})$ when trained on larger token budgets. This alignment between improved loss and higher effective rank helps explain the trends observed in §3.2, where models with larger $\mathcal{R}(\mathbf{W})$ tend to achieve lower loss.

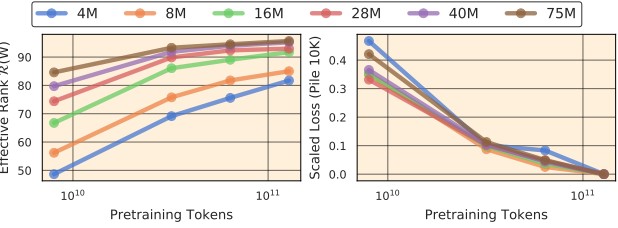

*Figure 12.* **Effect of token budget on $\mathcal{R}(\mathbf{W})$ and performance:** Training on more tokens yield better performance and higher effective rank across model sizes.

> **Takeways:** Pre-training on a larger token budget, with large batch size, sufficient weight decay, and less aggressive LR annealing can impede geometric collapse. However, this does not necessarily translate to a better in-distribution loss.

## 4.2. Limitations of Effective Rank as a Proxy for Performance Estimation

The mathematical formulation of the effective rank of the unembedding matrix, $\mathcal{R}(\mathbf{W})$, as defined in Equation 1, captures only the distributional spread of the singular value spectrum and does not necessarily reflect task-relevant information. As discussed above, this spectrum is known to be shaped by optimization dynamics. For example, large batch sizes average gradients over many tokens, mitigating frequency biases and preserving a broader spread of $\mathbf{W}$ across singular directions. However, excessive averaging at very large batch sizes can reduce the signal-to-noise ratio, hindering fine-grained learning. As a result, $\mathbf{W}$ may remain high-rank yet become informationally diffuse, resembling the behavior of randomly initialized models. Likewise, strong weight decay prevents any specific direction from dominating, thus increasing rank but reducing performance. Hence, we do not attribute a high effective rank to cause good performance. Instead, we find that well-performing models result from favourable hyperparameters, with high effective rank emerging as a byproduct. Moreover, rank suitability also depends on the task. Low rank might suffice for classification or clustering, where outputs are needed to concentrate in a few directions; whereas high rank may be suitable for open-ended tasks to support diverse outputs and capture the variation in the model's predictions. In summary, our empirical findings motivate a more rigorous theoretical analysis to characterize when a high effective rank in $\mathbf{W}$ enhances performance and when it does not.

## 5. Alternatives to Effective Rank

As mentioned in §2, prior work has also used cosine similarity and partition-function-based Isotropy as metrics for analyzing the geometry of the unembedding matrix (Machina & Mercer, 2024). While each metric purports to characterize model geometry, they capture fundamentally different properties. Effective rank is a magnitude-based direction-agnostic metric that quantifies how evenly variance is distributed across singular directions. On the other hand, Isotropy and cosine similarity are magnitude agnostic. The former answers how much more concentrated vectors are in their most-preferred direction vs. least-preferred, while the latter treats all vector pairs equally, rather than focusing on principal components and computes pairwise alignment. In this section, we study how these metrics correlate (or not) with each other for the unembedding matrix $\mathbf{W}$, and last-token's final layer representations $\mathbf{H} \in \mathbb{R}^{N \times d}$, where $N$ is the number of input sequences.

**Effective rank aligns with cosine similarity; Isotropy is primarily sensitive to weight decay.** Figure 13 presents the Pareto fronts of geometric metrics for the unembed-

ding matrix. Both cosine similarity $\mathcal{A}(\mathbf{W})$ and effective rank $\mathcal{R}(\mathbf{W})$ vary substantially across models and exhibit a non-monotonic, parabolic relationship. This '*moon-shaped*' correlation indicates that models with low rank embeddings tend to have high angular alignment, and vice-versa. However, this pattern is not strict: some high-rank models also exhibit substantial angular alignment. This is expected, as effective rank is direction-agnostic and cannot distinguish between different spatial arrangements that share the same singular value spectrum. In contrast, isotropy scores $\mathcal{I}(\mathbf{W})$ are tightly concentrated around 0.5 for most models, consistent with prior observations (Machina & Mercer, 2024). Across all model sizes, we see the ones trained with lower weight decay strength exhibit high anisotropy, but varying effective rank and high cosine similarity. We conjecture that this might be because of the unconstrained growth of the embeddings along task-relevant directions, causing vectors to concentrate in a few dominant directions that are exponentially amplified by the partition function.

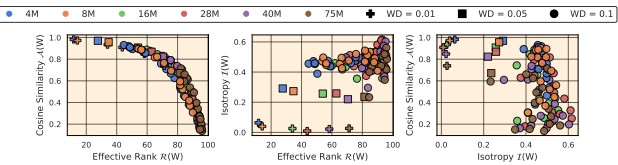

*Figure 13.* Pareto fronts of unembedding geometry metrics. Effective rank and cosine similarity follow a non-monotonic trend, while isotropy varies mainly with weight decay.

**Representation geometry varies less than the unembedding geometry.** From Figure 14, we observe that model representations have substantially higher rank and less variation than the unembedding matrix. They also exhibit low cosine similarity, aligning with prior observations (Machina & Mercer, 2024; Godey et al., 2024a; Razzhigaev et al., 2024). Interestingly, model representations with similar effective rank, $\mathcal{R}(\mathbf{H})$, span a wide variety of embedding similarity $\mathcal{A}(\mathbf{W})$. Similar trend is observed for $\mathcal{A}(\mathbf{H})$ vs $\mathcal{A}(\mathbf{W})$. This indicates that high alignment among embeddings does not directly translate to similarly aligned representations. The unembedding Isotropy, $\mathcal{I}(\mathbf{W})$, displays more or less similar trends when compared with either the model representations or the unembedding matrix. One notable exception is that models with low Isotropy in the unembedding matrix tend to produce representations with substantially lower cosine similarity. These models also correspond to training regimes with low weight decay. Lastly, we see that the geometry of $\mathbf{H}$ varies less than $\mathbf{W}$, as both are expected to capture different properties. The effective rank of $\mathbf{H}$ reflects variation

---

As per Arora et al. (2016), the partition function is useful to measure the global isotropy of a large, fixed set of vectors, such as $\mathbf{W}$. In contrast, $\mathbf{H}$ captures only a limited set of samples and cannot reflect the global geometry, making the metric unreliable for them. Hence, we compute it for $\mathbf{W}$ but not $\mathbf{H}$.

in final-layer representations across inputs, whereas **W** is directly shaped by gradients and encodes prediction errors. Since **H** arises from layered transformations and **W** from the output objective, there is no strong reason to expect their effective ranks to correlate. Thus, we conclude that they should not be analyzed interchangeably. We elaborate more on the effect of the last token's final-layer representation geometry on downstream performance in Appendix A.3.

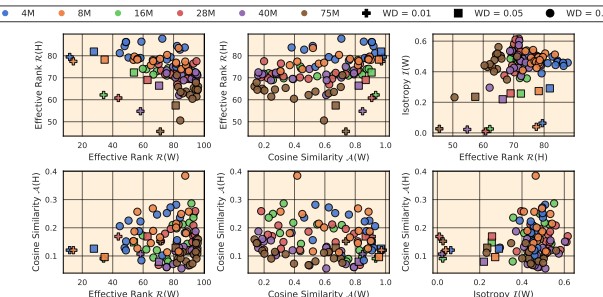

*Figure 14.* Pareto front of similarity and effective rank metrics for the unembedding matrix vs last layer representation. Representations show higher effective rank and less variation than the unembedding matrix, with low cosine similarity. Embedding's geometric variation does not directly transfer to representations.

## 6. Conclusion

In this work, we present a systematic study of the relationship between language model performance and the geometry of the unembedding matrix, particularly its effective rank. We train and analyze 108 pre-trained OLMo models spanning diverse training setups and observe that higher effective rank generally correlates with better performance. However, this relationship is neither necessary nor sufficient: across five evaluation tasks, we observe many counterexamples of high-rank models that underperform and low-rank models that do well. Revisiting the problem of saturation, i.e. performance degradation in late-stage pretraining, we find that low effective rank is a concurrent manifestation, not necessarily the root cause. Instead, effective rank strongly captures hyperparameter trends than model performance. Lastly, extending our analysis to cosine similarity, isotropy, and final-layer token representations, we find that these metrics broadly capture similar trends, and that representation geometry exhibits less variation than the unembedding matrix. Nevertheless, both fail to reliably predict downstream performance. Instead, performance is better improved by tuning standard hyperparameters (e.g., batch size and weight decay) rather than directly optimizing geometric properties. In summary, our results suggest that geometric metrics should be interpreted with caution and cannot serve as substitutes for downstream evaluation. We provide more detailed practical takeaways of our findings for ML practitioners and researchers in Appendix B and highlight the limitations of our work in Appendix C.

## Impact Statement

Our study focuses on understanding existing models rather than developing new capabilities, and our findings promote more careful interpretation of model geometry rather than enabling harmful applications. As such, we do not anticipate any direct negative societal impacts from this work. Our pre-training experiments require $\approx$ 3000 GPU hours. We aim to share the benefit of this cost by releasing all of our model checkpoints as open source. We encourage the research community to make use of them.

## Acknowledgements

This research is supported in part by the National Science Foundation (NSF) under grant IIS2403437, the Simons Foundation, Apple, Intel, Allen Institute for AI, and USC Women in Science and Engineering (WiSE) Gabilan Fellowship. Part of this research was done when Atharva Kulkarni and Swabha Swayamdipta were visitors at the Simons Institute for the Theory of Computing, UC Berkeley. Jacob Mitchell Springer was supported by NSF Graduate Research Fellowship under Grant No. DGE2140739. Any opinions, findings, conclusions, or recommendations expressed in this material are those of the author(s) and do not necessarily reflect the views of any of the funding organizations. The authors would like to thank Pierre Marion, Ian Magnusson, Amey Hengle, Matthew Finlayson, Gregory Yauney, and other members of the DILL lab.

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

# A. Appendix

## A.1. Experimental Details

**Pre-Training Setup**   As mentioned in Section 3.1, we pre-train a suite of OLMo-style (Groeneveld et al., 2024) models using the model configurations of Magnusson et al. (2025). Specifically, all models have 6 layers, 4 attention heads, an MLP ratio of 4, SwiGLU activation, a sequence length of 2048 and a vocabulary of 50304 tokens. We use RMS LayerNorm (Zhang & Sennrich, 2019) across all layers and do not use any dropout. Table 1 details the hidden dimension for all the model sizes and the model-specific learning rates inferred from the Porian et al. (2024)'s scaling laws. We use the raw Pile dataset (Gao et al., 2020) for pre-training these models, which is tokenized using the allenai/gpt-neox-olmo-dolma-v1_5 tokenizer (Soldaini et al., 2024).

| Hyperparameters | OLMo-4M | OLMo-8M | OLMo-16M | OLMo-28M | OLMo-40M | OLMo-75M |
|---|---|---|---|---|---|---|
| Hidden dimensions | 64 | 128 | 256 | 384 | 512 | 768 |
| Optimizer | AdamW | AdamW | AdamW | AdamW | AdamW | AdamW |
| $\beta_1$ | 0.9 | 0.9 | 0.9 | 0.9 | 0.9 | 0.9 |
| $\beta_2$ | 0.95 | 0.95 | 0.95 | 0.95 | 0.95 | 0.95 |
| Learning Rate | 1.5e-2 | 1.1e-2 | 8.7e-3 | 7.4e-3 | 6.5e-3 | 5.3e-3 |
| Weight decay | 0.1 | 0.1 | 0.1 | 0.1 | 0.1 | 0.1 |
| LR Scheduler | Cosine | Cosine | Cosine | Cosine | Cosine | Cosine |
| LR Annealing | 10% | 10% | 10% | 10% | 10% | 10% |
| Warmup | 10% | 10% | 10% | 10% | 10% | 10% |

*Table 1.* Pre-training hyperparameters used in our controlled experiments.

Table 2 details the choice of hyperparameters in our suite. It is important to note that we do not train models on extremely larger token budget + smaller batch sizes as they require a high number of optimization steps (eg: training 128B tokens with batch size 32 needs 2M steps to finish training), necessitating heavy GPU requirements. Thus, for large token budgets, we chose batch sizes such that the optimization steps do not exceed 150k steps.

| Hyperparameter | Tokens | Batch Size | Weight Decay | Learning Rate | LR Decay | Total |
|---|---|---|---|---|---|---|
| Batch Size | 8B | 32, 64, 128, 256, 512 | 0.1 | Model specific | 1% | 5 |
| Weight decay | 8B | 32 | 0.1, 0.5, 0.01 | Model specific | 1% | 3 |
| Learning rate | 8B | 32 | 0.1 | $[1, 0.1, 10] \times$ Model specific | 1% | 3 |
| LR Decay | 8B | 32 | 0.1 | Model specific | 0, 1%, 10% | 3 |
| Token budget | 8B, 32B, 64B, 128B | 32, 64, 125, 512 resp | 0.1 | Model specific | 1% | 4 |

*Table 2.* Hyperparameter details for our suite. Each model has 18 variants and we train for 6 model sizes, resulting in 108 models.

**Fine-Tuning Setup**   We fine-tune the last checkpoints of all our models on the StarCoder-Python (Li et al., 2023) and OpenWebMath (Paster et al., 2024) datasets. We use a batch size of 32 and 10% smaller learning rate than the model-specific learning rates for fine-tuning, keeping all other hyperparameters the same.

## A.2. Extended Discussion: The Effect of Hyperparamters on Performance and the Unembedding / Representation Geometry

### A.2.1. EFFECT OF BATCH SIZE.

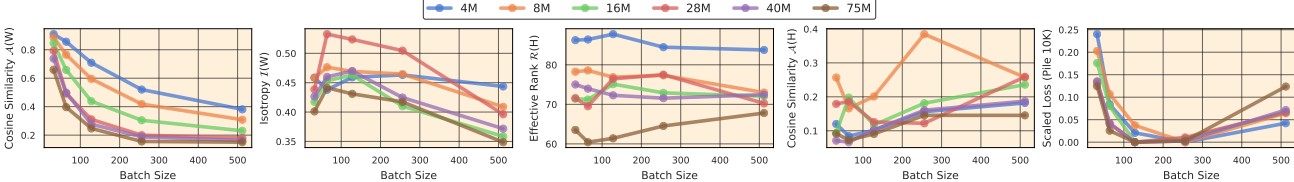

*Figure 15.* Effect of batch size on in-distribution performance and different geometric metrics for unembedding matrix as well as final-token's last-layer representations.

Figure 15 shows how batch size affects the geometric properties of learned representations and their relationship to model performance. As established in Section 4, scaled loss exhibits a U-shaped relationship with batch size, initially decreasing before eventually degrading at very large batch sizes. The isotropy of the unembedding matrix, $\mathcal{I}(\mathbf{W})$, demonstrates strong visual correlation with this performance trend, displaying a corresponding inverted U-shaped curve that mirrors the loss dynamics across all model sizes. In contrast, the cosine similarity of the unembedding matrix, $\mathcal{A}(\mathbf{H})$, shows a monotonic decline with increasing batch size, suggesting progressive orthogonalization of output embeddings that does not directly track performance. Finally, the effective rank $\mathcal{R}(\mathbf{H})$ and cosine similarity $\mathcal{A}(\mathbf{H})$ of the final token's last layer representations exhibit minimal variation across different batch sizes and fail to capture the performance dynamics.

### A.2.2. EFFECT OF WEIGHT DECAY.

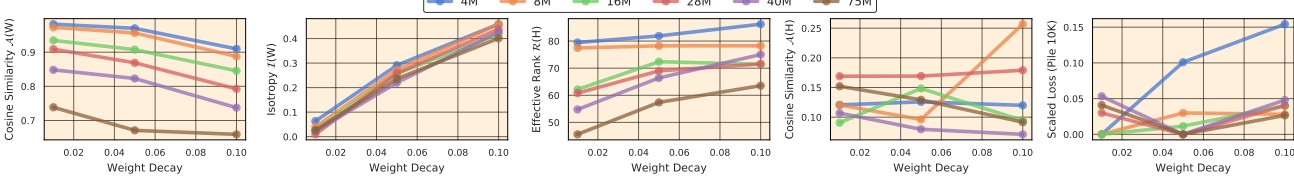

*Figure 16.* Effect of weight decay on in-distribution performance and different geometric metrics for unembedding matrix as well as final-token's last-layer representations.

The impact of weight decay on in-distribution loss is model size dependent, as elaborated in Section 3. From Figure 16, we observe that the isotropy $\mathcal{I}(\mathbf{W})$ exhibits a steady increase with stronger weight decay regularization. Similarly, the cosine similarity among the unembedding vectors $\mathcal{A}(\mathbf{W})$ decreases monotonically with increasing weight decay strength. Both trends suggest that strong weight decay prevents the model from clustering representations along any particular direction, promoting more uniform distribution in the embedding space. While the cosine similarity of the final layer representations $\mathcal{A}(\mathbf{H})$ remains nearly constant across weight decay values (with a minor exception for the OLMo-8M model), the effective rank $\mathcal{R}(\mathbf{H})$ increases under high weight decay. Nevertheless, none of these geometric metrics fully capture the complex, performance trends induced by weight decay, indicating that the relationship between regularization-induced geometric changes and loss is more nuanced than for batch size effects.

### A.2.3. EFFECT OF LEARNING RATE.

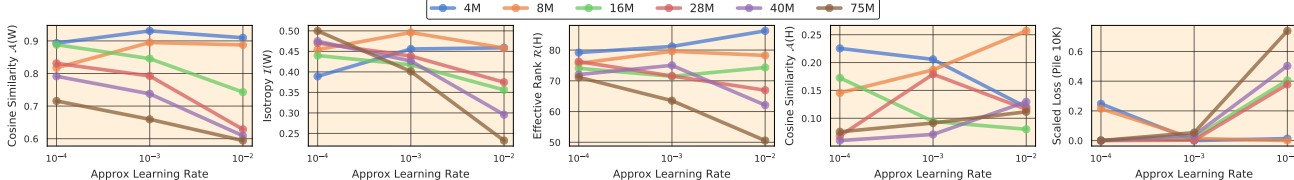

*Figure 17.* Effect of learning rate on in-distribution performance and different geometric metrics for unembedding matrix as well as final-token's last-layer representations.

As seen in Figure 17, the learning rate has an inconsistent, model-size-dependent impact on performance. Its effect on

unembedding and representation geometry is inconsistent, and neither metric seems to capture the performance trend induced by the learning rate.

### A.2.4. EFFECT OF LEARNING RATE ANNEALING.

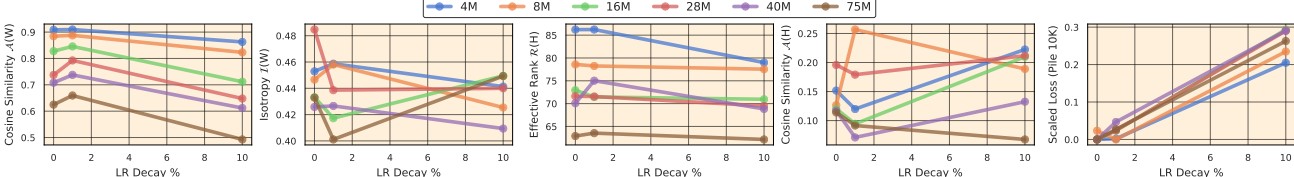

*Figure 18.* Effect of learning rate annealing on in-distribution performance and different geometric metrics for unembedding matrix as well as final-token's last-layer representations.

Annealing the learning rate to a higher minimum value consistently results in higher loss compared to annealing to zero, as evidenced in Figure 18. This is plausible as, without fully annealing the learning rate (keep it at some non-zero value), the model continues taking optimization steps and can't settle into a good minimum, leading to higher loss. The steps become larger as the annealing percentage increases. The cosine similarity of the unembedding matrix $\mathcal{A}(\mathbf{W})$ captures this performance degradation, showing a substantial decrease as the final learning rate increases, suggesting that the representations are converging in the embedding space (a sign of overfitting). The effective rank of the final layer representations $\mathcal{R}(\mathbf{H})$ follows a similar decline. However, the cosine similarity of representations $\mathcal{A}(\mathbf{H})$ and the isotropy of the unembedding matrix $\mathcal{I}(\mathbf{W})$ exhibit inconsistent trends across different final learning rates, failing to reliably track the performance changes induced by learning rate annealing.

### A.2.5. EFFECT OF PRETRAINING TOKEN BUDGET.

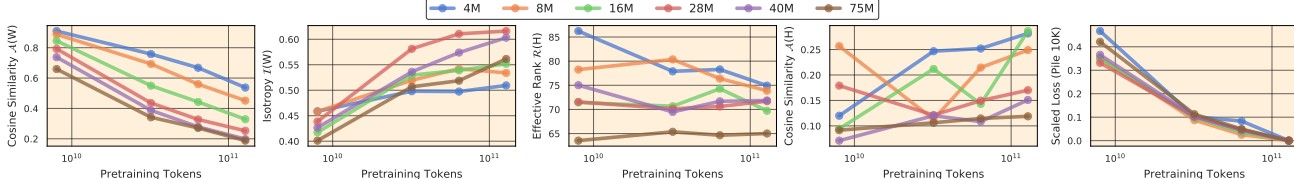

*Figure 19.* Effect of token budget on in-distribution performance and different geometric metrics for unembedding matrix as well as final-token's last-layer representations.

Results in Figure 19 support the well-established finding that training on more tokens improves model performance. The cosine similarity of the unembedding matrix $\mathcal{A}(\mathbf{W})$ correlates well with this loss trend, exhibiting consistent decline as the pretraining token budget increases. Similarly, the isotropy $\mathcal{I}(\mathbf{W})$ shows negative correlation with loss, increasing monotonically with larger token budgets. Both metrics indicate that extended training promotes more orthogonal and uniformly distributed unembedding vectors. However, the geometric metrics of the final layer representations – $\mathcal{A}(\mathbf{H})$ and $\mathcal{R}(\mathbf{H})$ – exhibit inconsistent trends across token budgets, rendering them unsuitable for predicting performance improvements from increased training data.

## A.3. Extended Discussion: The Impact of Model Geometry on Downstream Performance

In this section, we evaluate how well geometric metrics – particularly effective rank, cosine similarity, and isotropy – computed from the unembedding matrix $\mathbf{W}$ and the final-token last-layer representations $\mathbf{H}$ predict downstream performance across a range of tasks. In addition to scatter plots for qualitative analysis, we quantify these relationships using the following complementary regression-based measures:

- **Raw Spearman correlation:** Measures the unadjusted monotonic relationship between each geometric metric and loss without accounting for model size or any training hyperparameter.

- **Residual Spearman (linear):** Computes the Spearman correlation between each metric and loss residuals after linearly regressing out model size and token budget.

- **Residual Spearman (scaling-law):** Uses residuals obtained by fitting Chinchilla scaling laws (Hoffmann et al., 2022), followed by Spearman correlation with each metric.

- **Partial Spearman correlation:** Assesses the unique association between each geometric metric and loss while controlling for all training hyperparameters, providing the most stringent test of predictive reliability.

- **Predictive $\Delta R^2$:** Measures the additional explained variance contributed by each geometric metric beyond a baseline model, including model size, compute, and token count, after a K-FOLD validation (k=5).

A.3.1. IN-DISTRIBUTION EVALUATION.

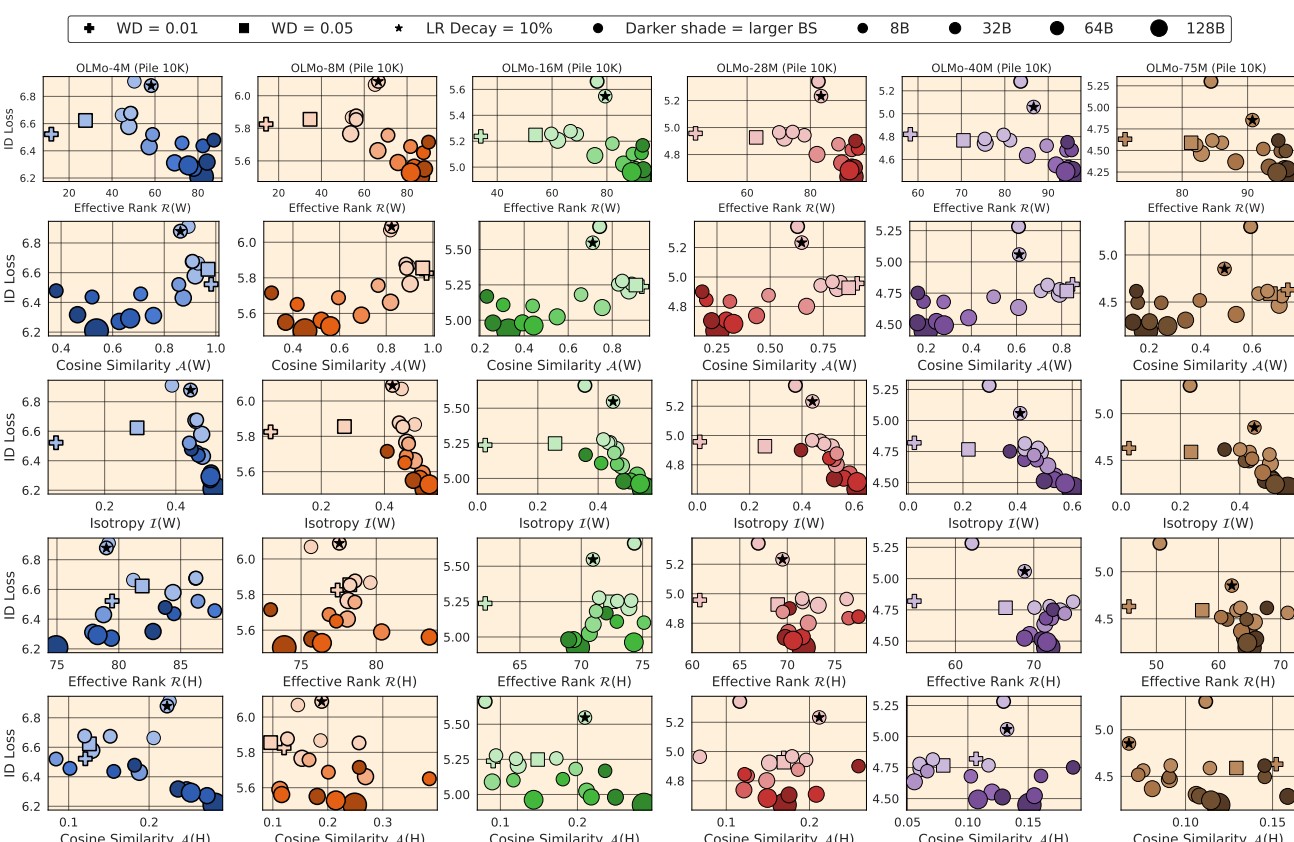

*Figure 20.* Extended in-distribution evaluation in relation to geometric metrics of the unembedding matrix and last-layer final-token representation.

The cosine similarity of the unembedding matrix, $\mathcal{A}(\mathbf{W})$, depicts a U-shaped trend with in-domain loss across model sizes, suggesting that higher angular alignment does not always yield the best performance. These are the models trained on smaller token budgets with higher batch sizes, which may hint that they did not converge due to fewer steps. As evidenced

| Analysis | Effective Rank $\mathcal{R}(\mathbf{W})$ | Cosine Similarity $\mathcal{A}(\mathbf{W})$ | Isotropy $\mathcal{I}(\mathbf{W})$ | Effective Rank $\mathcal{R}(\mathbf{H})$ | Cosine Similarity $\mathcal{A}(\mathbf{H})$ |
|---|---|---|---|---|---|
| Raw Spearman $\rho$ | -0.699* | 0.601* | -0.241* | 0.773* | 0.388* |
| Residual Spearman (linear) | -0.209* | 0.233* | -0.155 | -0.008 | -0.174 |
| Residual Spearman (Hoffmann) | -0.724* | 0.635* | -0.333* | 0.729* | 0.337* |
| Partial Spearman | -0.130 | -0.170 | -0.280* | -0.122 | -0.033 |
| Predictive $\Delta R^2$ | -0.003 | 0.001 | 0.029 | 0.009 | -0.001 |

*Table 3.* ID Loss (Pile 10K) correlation analysis summary. Values marked with ∗ are significant with $p < 0.05$.

before in Section 5, Isotropy $\mathcal{A}(\mathbf{W})$ shows a sharp decline when trained with lower weight decay, but does not have any impact on the ID loss. The effective rank of the representations $\mathcal{R}(H)$ remains largely constant across different model sizes, thus unable to capture performance trends. While the cosine similarity of the representations $\mathcal{A}(\mathbf{H})$ is generally lower and decreases with increasing model size, it cannot predict intra-model performance differences across different training setups. The regression analysis results in Table 3 support our conclusions. Raw Spearman correlations show that $\mathcal{R}(\mathbf{H})$ (effective rank of representations) has the strongest correlation with loss (0.773), followed by $\mathcal{R}(\mathbf{W})$ (−0.699) and $\mathcal{A}(\mathbf{W})$ (0.601). However, these raw correlations are heavily confounded by model size and token budget. When controlling for a simple linear scaling law (Residual Spearman linear), all correlations drop dramatically (ranging from −0.209 to 0.233) More importantly, when controlling for the Chinchilla scaling law (Residual Spearman Chinchilla), correlations remain substantial for $\mathcal{R}(\mathbf{W})$ (−0.736), $\mathcal{A}(\mathbf{W})$ (0.656), and $\mathcal{R}(\mathbf{H})$ (0.669), indicating these metrics capture loss variance *not* explained by the model size and token budgets. However, Partial Spearman correlations, which control for training hyperparameters simultaneously, show very weak values, suggesting that these metrics are extremely weak predictors of in-distribution loss. Only $\mathcal{I}(\mathbf{W})$ shows a moderate correlation of −0.337, which cannot be considered reliable. The predictive $\Delta R^2$ values are extremely small (0.002–0.004), indicating that despite residual correlations with Chinchilla-adjusted loss, these geometric metrics provide negligible additional predictive power (< 0.5% variance explained) beyond existing scaling laws.

A.3.2. OUT-OF-DISTRIBUTION EVALUATION.

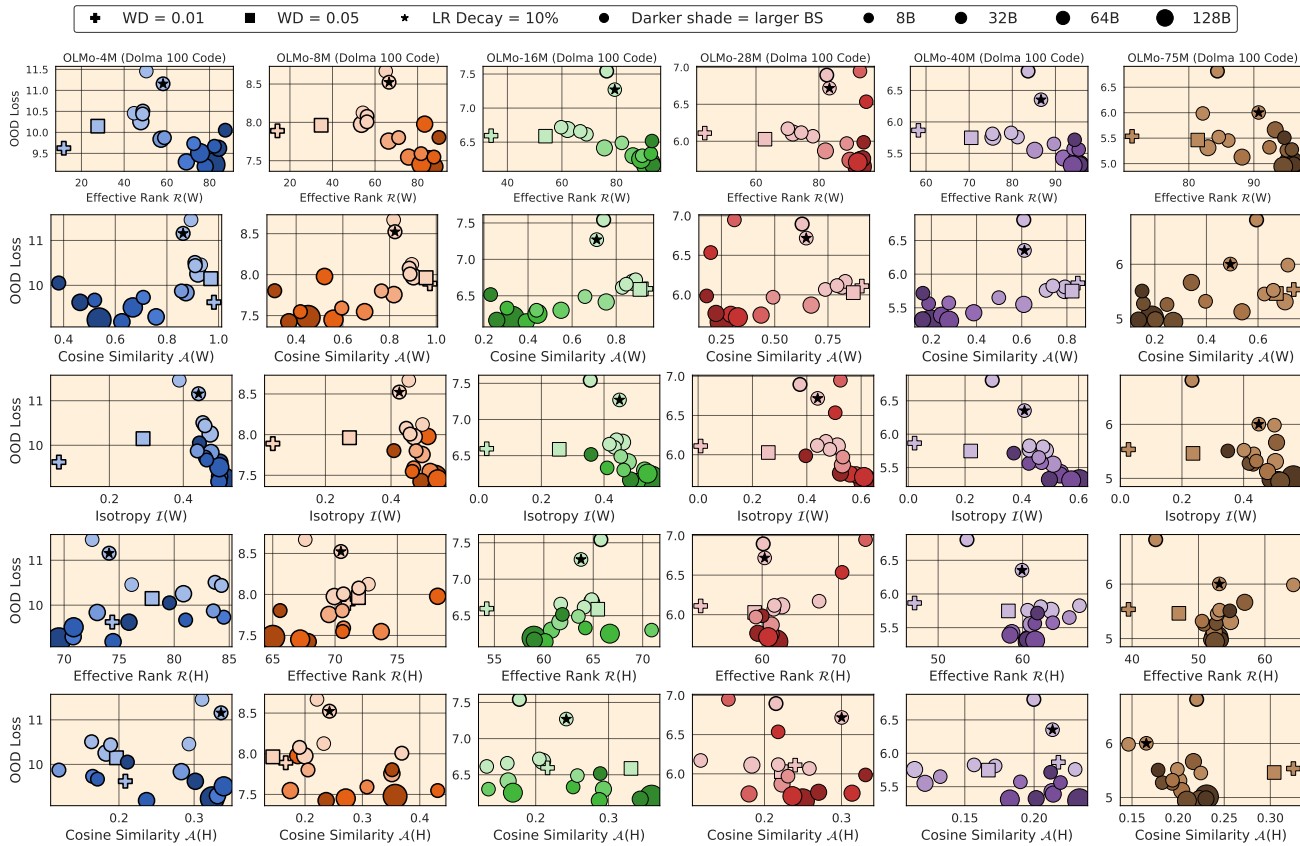

*Figure 21.* Extended out-of-distribution evaluation in relation to geometric metrics of the unembedding matrix and last-layer final-token representation.

| Analysis | Effective Rank $\mathcal{R}(\mathbf{W})$ | Cosine Similarity $\mathcal{A}(\mathbf{W})$ | Isotropy $\mathcal{I}(\mathbf{W})$ | Effective Rank $\mathcal{R}(\mathbf{H})$ | Cosine Similarity $\mathcal{A}(\mathbf{H})$ |
|---|---|---|---|---|---|
| Raw Spearman $\rho$ | -0.672* | 0.579* | -0.178 | 0.831* | 0.131 |
| Residual Spearman (linear) | -0.038 | 0.065 | -0.040 | 0.002 | -0.116 |
| Residual Spearman (Hoffman) | -0.693* | 0.609* | -0.249* | 0.798* | 0.094 |
| Partial Spearman | -0.127 | -0.226* | -0.162 | 0.117 | -0.028 |
| Predictive $\Delta R^2$ | -0.004 | 0.010 | 0.020 | -0.009 | -0.004 |

*Table 4.* OOD Loss (Dolma 100 Code) correlation analysis summary. Values marked with $*$ are significant with $p < 0.05$.

Out-of-distribution results in Figure 21 largely mirror the in-distribution trends; therefore, we omit a detailed discussion of the scatter plots. The regression results in Table 4 are similarly consistent with the in-distribution analysis. Specifically, correlations decrease across all metrics and analysis settings, except for a marginal increase in $\mathcal{R}(\mathbf{H})$. Moreover, unlike the ID evaluation $\mathcal{I}(\mathbf{W})$, the Partial Spearman correlation drops during OOD evaluation, suggesting that it's an unreliable metric for performance evaluation. Overall, these findings align with the conclusions of Section 3: out-of-distribution performance is better predicted by in-distribution loss than by geometric metrics.

### A.3.3. POST TRAINING QUANTIZATION EVALUATION.

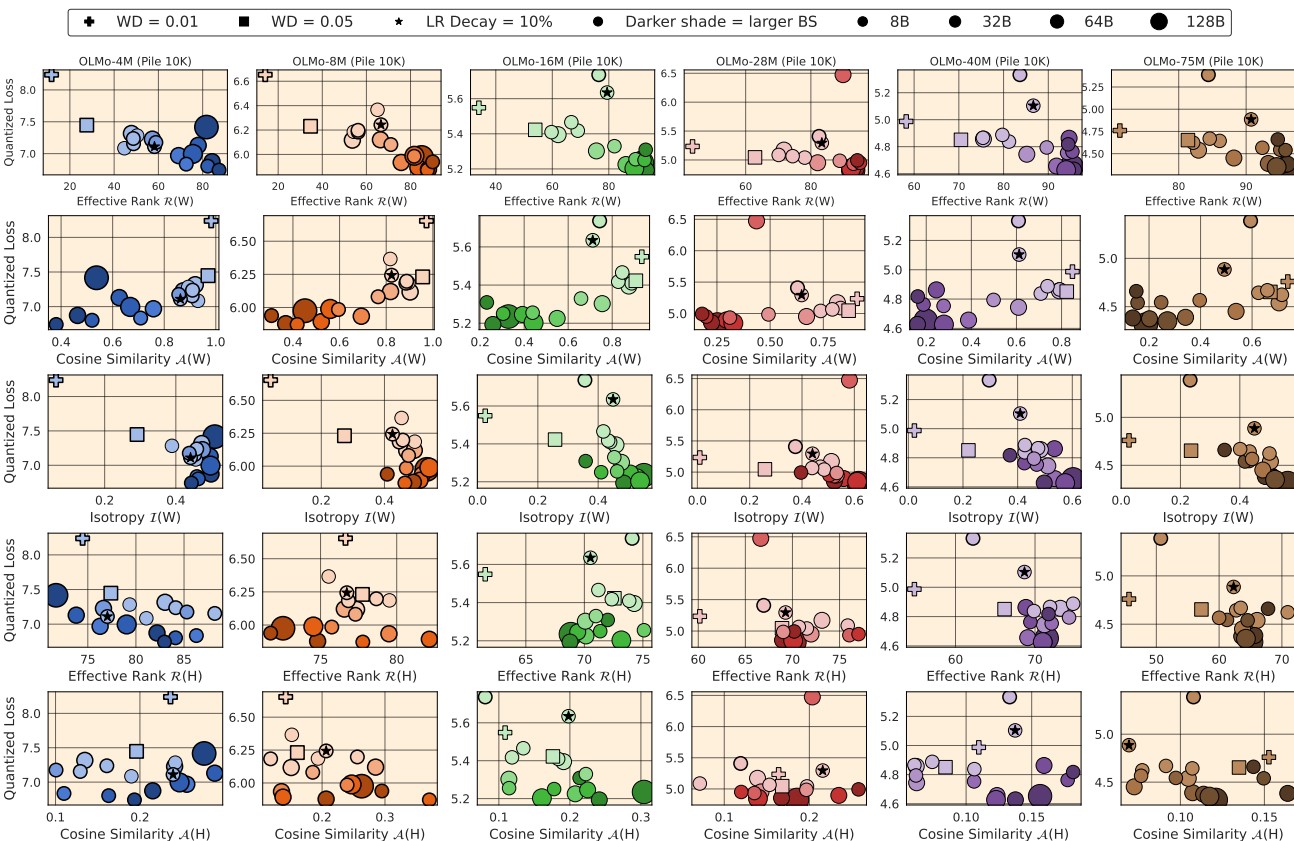

*Figure 22.* Extended post-training quantization evaluation on in-domain data in relation to geometric metrics of the unembedding matrix and last-layer final-token representation.

We analyze the sensitivity of OLMo models to post-training quantization (PTQ) on both in-domain and out-of-domain data. Figure 22 shows that, across all geometric metrics, models trained with weight decay 0.01 perform consistently worse under PTQ for the smaller OLMo-4M and OLMo-8M models. Conversely, the least impacted models across sizes tend to exhibit lower effective rank $\mathcal{R}(\mathbf{W})$, lower cosine similarity $\mathcal{A}(\mathbf{W})$, and higher isotropy $\mathcal{I}(\mathbf{W})$. In contrast, representation-level metrics, $\mathcal{R}(\mathbf{H})$ and $\mathcal{A}(\mathbf{H})$, do not display consistent trends: $\mathcal{R}(\mathbf{H})$ remains largely invariant, while $\mathcal{A}(\mathbf{H})$ spans a wide range among well-performing models. The regression results in Table 5 follow the expected pattern, with high raw and Chinchilla-adjusted Spearman correlations but weak partial correlations and negligible predictive $\Delta R^2$. Together, these findings indicate that most variance in PTQ robustness is not explained by the considered geometric metrics, suggesting the influence of additional underlying mechanisms. Results for out-of-domain PTQ in Figure 23 and Table 6 exhibit similar

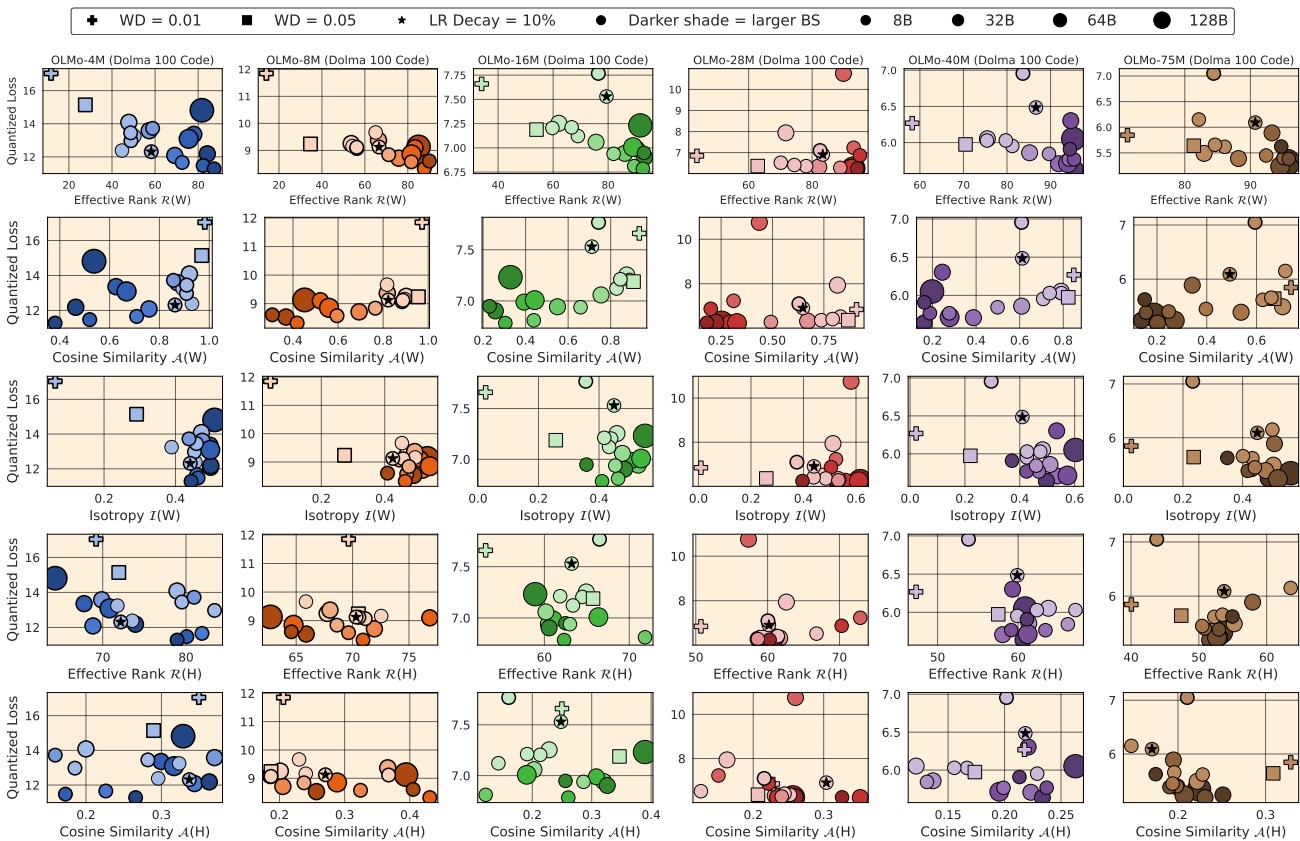

*Figure 23.* Extended post-training quantization evaluation on out-of-domain data in relation to geometric metrics of the unembedding matrix and last-layer final-token representation.

| Analysis | Effective Rank $\mathcal{R}(\mathbf{W})$ | Cosine Similarity $\mathcal{A}(\mathbf{W})$ | Isotropy $\mathcal{I}(\mathbf{W})$ | Effective Rank $\mathcal{R}(\mathbf{H})$ | Cosine Similarity $\mathcal{A}(\mathbf{H})$ |
|---|---|---|---|---|---|
| Raw Spearman $\rho$ | -0.687* | 0.585* | -0.163 | 0.726* | 0.517* |
| Residual Spearman (linear) | -0.229* | 0.259* | -0.230* | -0.135 | -0.209* |
| Residual Spearman (Hoffmann) | -0.709* | 0.613* | -0.228* | 0.676* | 0.477* |
| Partial Spearman | -0.210* | -0.174 | -0.151 | -0.069 | -0.066 |
| Predictive $\Delta R^2$ | 0.006 | 0.001 | -0.000 | 0.004 | -0.003 |

*Table 5.* PTQ Loss (Pile 10K) correlation analysis summary. Values marked with * are significant with $p < 0.05$.

trends, further reinforcing our conclusions about the limited explanatory power of geometric metrics for PTQ behavior.

### A.3.4. FINE-TUNING EVALUATION.

The fine-tuning scatter plots for StarCoder-Python and OpenWebMath are shown in Figures 24 and 25, respectively. For smaller models (4M-16M), the unembedding metrics $\mathcal{R}(\mathbf{W})$ and $\mathcal{A}(\mathbf{W})$ exhibit substantial variance, while representation metrics $\mathcal{A}(\mathbf{H})$ and $\mathcal{R}(\mathbf{H})$ are tightly clustered. In both cases, these behaviors indicate limited predictive value for fine-tuning performance. For the larger models (28M-75M), lower effective rank $\mathcal{R}(\mathbf{W})$ and higher cosine similarity $\mathcal{A}(\mathbf{W})$ are more frequently associated with better performance, though this trend is neither strong nor consistent. Isotropy $\mathcal{I}(\mathbf{W})$ shows a weak linear relationship with fine-tuning loss for larger models, but this pattern does not hold for smaller models. Supporting regression analyses in Tables 7 and 8 confirm these observations, with most metrics exhibiting weak partial Spearman correlations. Although $\mathcal{R}(\mathbf{H})$ surprisingly, shows a relatively strong association for StarCoder-Python, this effect does not generalize to OpenWebMath. Overall, none of the considered geometric metrics consistently predict fine-tuning performance across both datasets.

| Analysis | Effective Rank $\mathcal{R}(\mathbf{W})$ | Cosine Similarity $\mathcal{A}(\mathbf{W})$ | Isotropy $\mathcal{I}(\mathbf{W})$ | Effective Rank $\mathcal{R}(\mathbf{H})$ | Cosine Similarity $\mathcal{A}(\mathbf{H})$ |
|---|---|---|---|---|---|
| Raw Spearman $\rho$ | -0.659* | 0.559* | -0.058 | 0.772* | 0.336* |
| Residual Spearman (linear) | -0.071 | 0.110 | -0.150 | -0.155 | -0.144 |
| Residual Spearman (Hoffmann) | -0.664* | 0.573* | -0.094 | 0.731* | 0.308* |
| Partial Spearman | -0.223* | -0.192* | -0.066 | 0.058 | 0.036 |
| Predictive $\Delta R^2$ | 0.018 | 0.001 | -0.008 | -0.003 | -0.009 |

*Table 6.* PTQ Loss (Dolma 100 Code) correlation analysis summary. Values marked with ∗ are significant with $p < 0.05$.

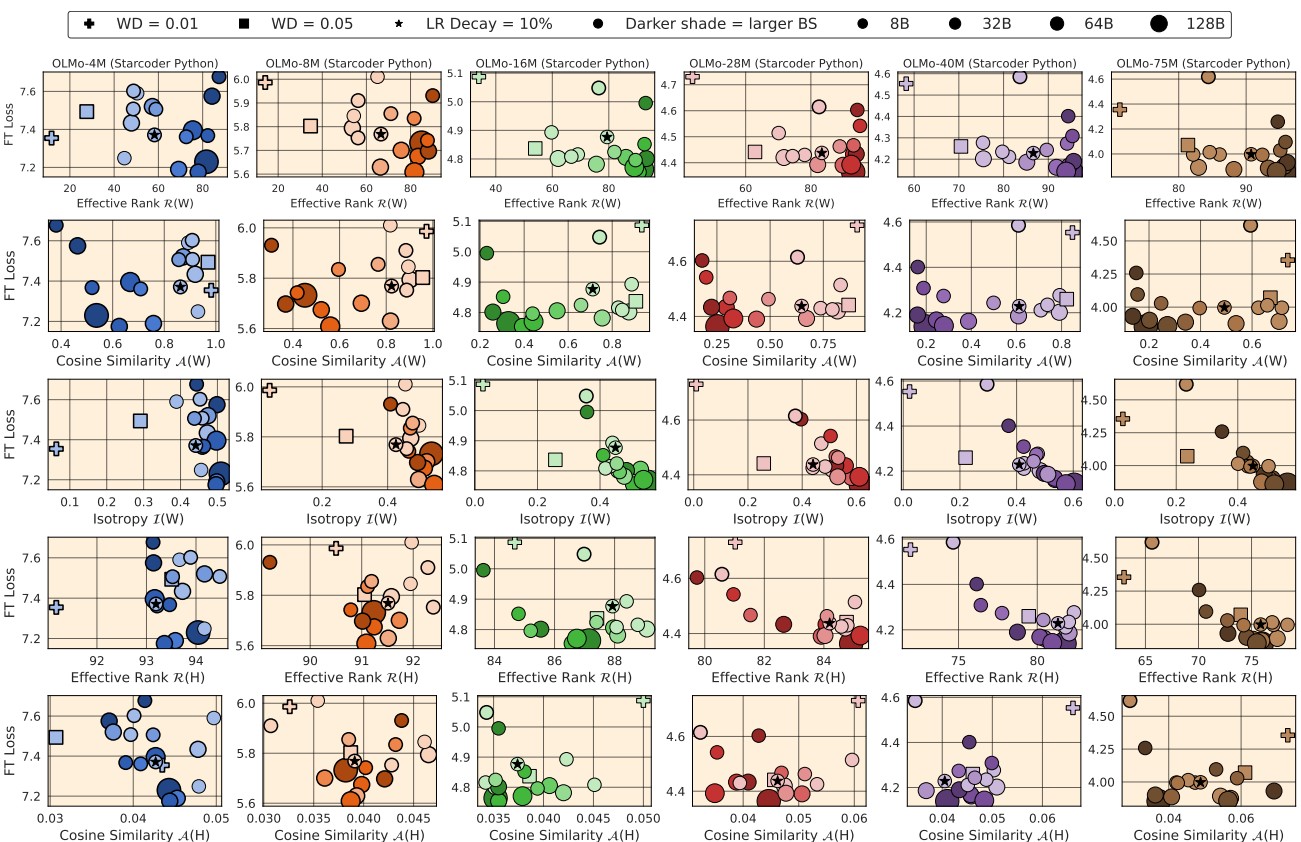

*Figure 24.* Extended fine-tuning evaluation (StarCoder-Python) in relation to geometric metrics of the unembedding matrix and last-layer final-token representation.

### A.3.5. CATASTROPHIC FORGETTING EVALUATION.

Figures 27 and 27 show in-distribution loss on Pile-10k for models fine-tuned on StarCoder-Python and OpenWebMath, respectively. Across model sizes, both the effective rank and the cosine similarity computed for the unembedding matrix, as well as the last-token final-layer representations, remain largely constant and therefore provide little signal for predicting catastrophic forgetting. Interestingly, isotropy $\mathcal{I}(\mathbf{W})$ is the only metric that exhibits a weak linear relationship with post–fine-tuning loss. However, models trained with weight decay 0.01 form a notable exception, displaying very low isotropy while still achieving strong performance. Regression results in Tables 9 and 10 corroborate these observations, indicating that catastrophic forgetting is largely invariant to the base model's unembedding and representation geometry.

| Analysis | Effective Rank $\mathcal{R}(\mathbf{W})$ | Cosine Similarity $\mathcal{A}(\mathbf{W})$ | Isotropy $\mathcal{I}(\mathbf{W})$ | Effective Rank $\mathcal{R}(\mathbf{H})$ | Cosine Similarity $\mathcal{A}(\mathbf{H})$ |
|---|---|---|---|---|---|
| Raw Spearman $\rho$ | -0.629* | 0.510* | -0.163 | 0.907* | -0.358* |
| Residual Spearman (linear) | 0.044 | -0.054 | -0.120 | -0.142 | 0.265* |
| Residual Spearman (Hoffmann) | -0.633* | 0.509* | -0.225* | 0.865* | -0.348* |
| Mixed-effects coef. | -0.000 | 0.017 | -1.359* | -0.037* | -3.496* |
| Predictive $\Delta R^2$ | -0.003 | -0.003 | 0.013 | 0.071 | -0.010 |

*Table 7.* Fine-tuning Loss (Starcoder Python) correlation analysis summary. Values marked with $*$ are significant with $p < 0.05$.

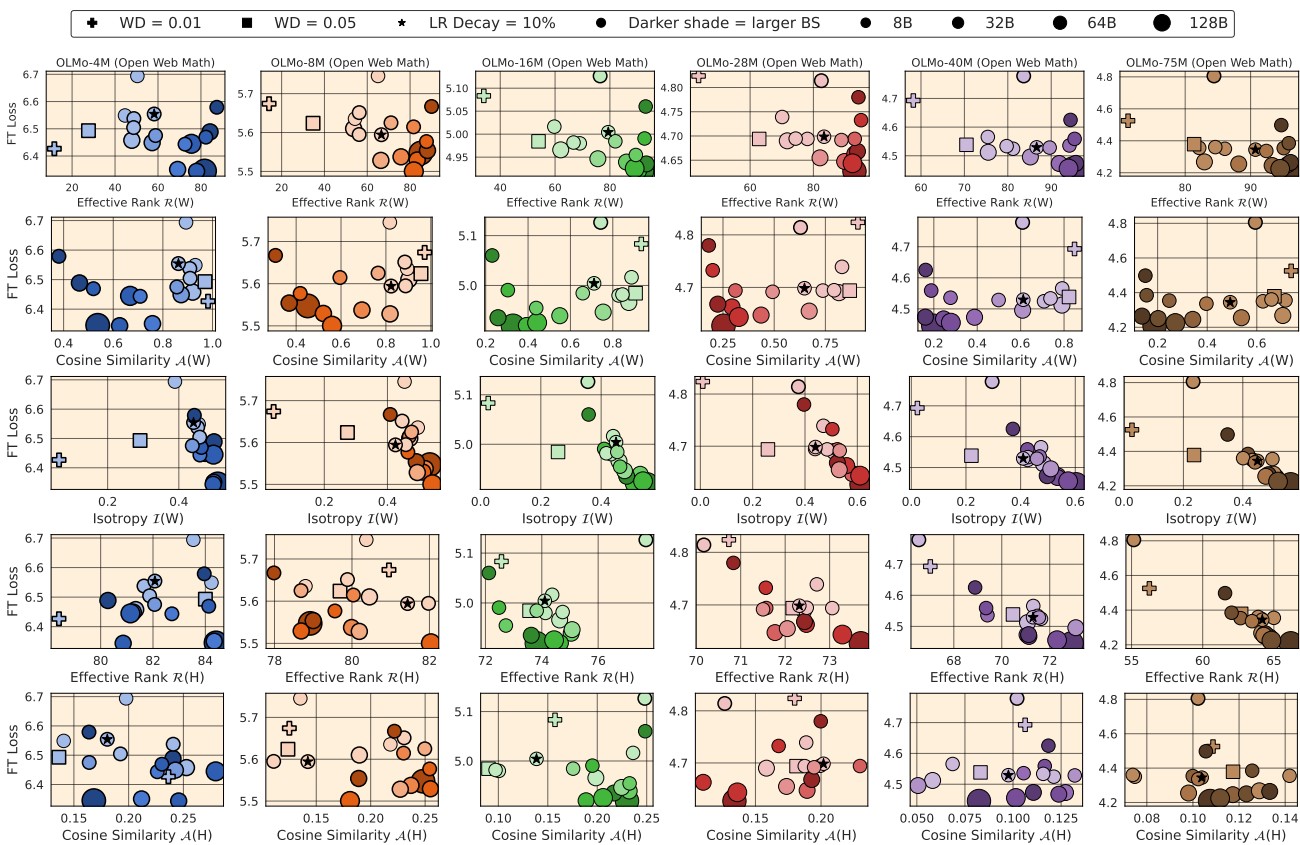

*Figure 25.* Extended fine-tuning evaluation (OpenWebMath) in relation to geometric metrics of the unembedding matrix and last-layer final-token representation.

| Analysis | Effective Rank $\mathcal{R}(\mathbf{W})$ | Cosine Similarity $\mathcal{A}(\mathbf{W})$ | Isotropy $\mathcal{I}(\mathbf{W})$ | Effective Rank $\mathcal{R}(\mathbf{H})$ | Cosine Similarity $\mathcal{A}(\mathbf{H})$ |
|---|---|---|---|---|---|
| Raw Spearman $\rho$ | -0.642* | 0.526* | -0.163 | 0.905* | 0.651* |
| Residual Spearman (linear) | 0.008 | -0.021 | -0.114 | -0.111 | -0.245* |
| Residual Spearman (Hoffmann) | -0.649* | 0.530* | -0.208* | 0.875* | 0.632* |
| Partial Spearman | -0.177 | -0.060 | -0.187 | -0.308* | -0.333* |
| Predictive $\Delta R^2$ | -0.002 | -0.002 | 0.015 | 0.030 | 0.009 |

*Table 8.* Fine-tuning Loss (Open Web Math) correlation analysis summary. Values marked with $*$ are significant with $p < 0.05$.

| Analysis | Effective Rank $\mathcal{R}(\mathbf{W})$ | Cosine Similarity $\mathcal{A}(\mathbf{W})$ | Isotropy $\mathcal{I}(\mathbf{W})$ | Effective Rank $\mathcal{R}(\mathbf{H})$ | Cosine Similarity $\mathcal{A}(\mathbf{H})$ |
|---|---|---|---|---|---|
| Raw Spearman $\rho$ | -0.591* | 0.486* | 0.024 | 0.808* | 0.685* |
| Residual Spearman (linear) | 0.024 | 0.001 | 0.045 | -0.041 | 0.048 |
| Residual Spearman (Hoffmann) | -0.580* | 0.479* | 0.029 | 0.785* | 0.681* |
| Partial Spearman | -0.176 | 0.017 | -0.154 | 0.023 | 0.274* |
| Mixed-effects coef. | -0.002 | 0.256 | -0.927* | -0.012* | 0.218 |
| Predictive $\Delta R^2$ | 0.001 | -0.002 | 0.007 | -0.000 | 0.004 |

*Table 9.* Post fine-tuning (Starcoder Python) in-domain Loss (Pile-10K) correlation analysis summary. Values marked with $*$ are significant with $p < 0.05$.

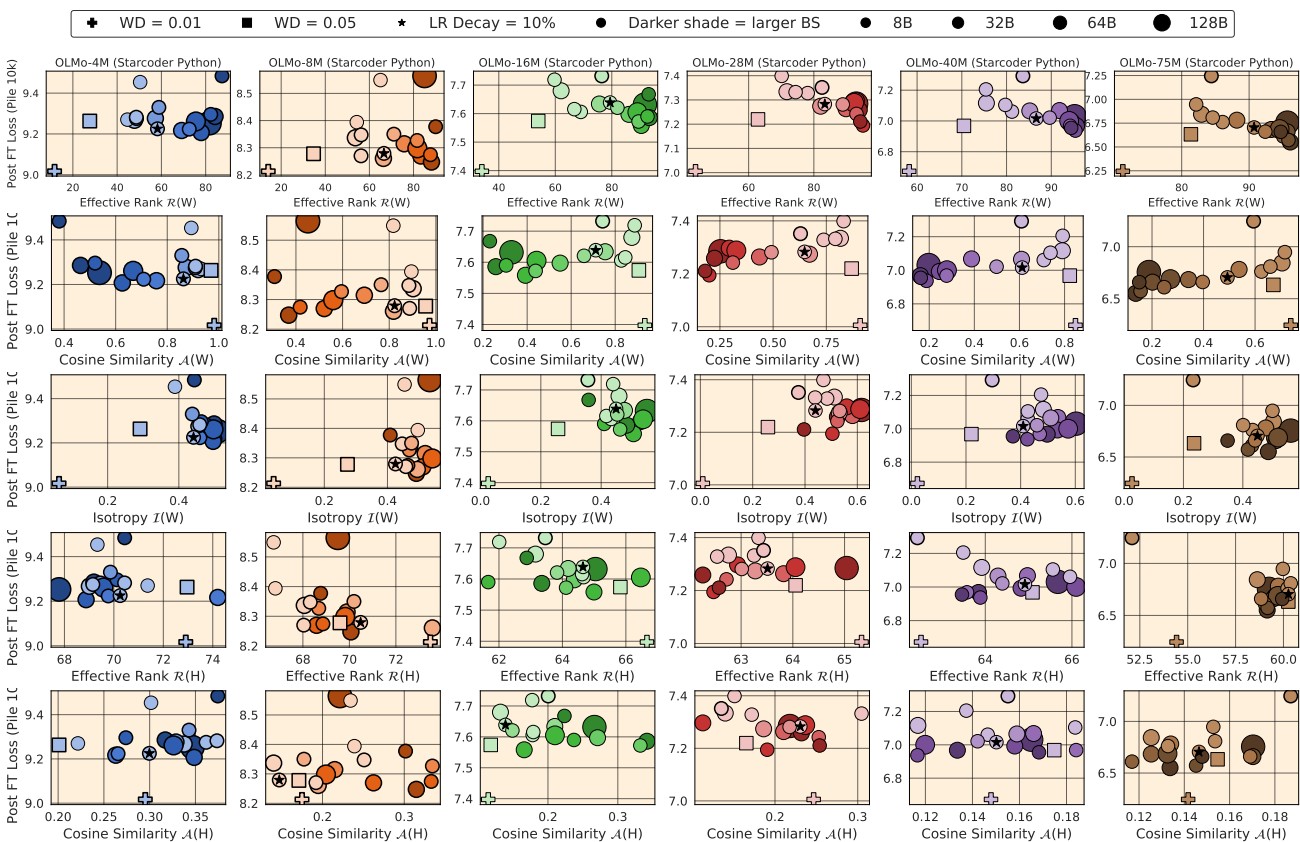

*Figure 26.* Extended forgetting evaluation (StarCoder-Python) in relation to geometric metrics of the unembedding matrix and last-layer final-token representation.

| Analysis | Effective Rank $\mathcal{R}(\mathbf{W})$ | Cosine Similarity $\mathcal{A}(\mathbf{W})$ | Isotropy $\mathcal{I}(\mathbf{W})$ | Effective Rank $\mathcal{R}(\mathbf{H})$ | Cosine Similarity $\mathcal{A}(\mathbf{H})$ |
|---|---|---|---|---|---|
| Raw Spearman $\rho$ | -0.626* | 0.519* | -0.083 | 0.877* | 0.640* |
| Residual Spearman (linear) | -0.025 | 0.030 | -0.029 | -0.006 | -0.204* |
| Residual Spearman (Hoffmann) | -0.629* | 0.523* | -0.092 | 0.863* | 0.632* |
| Partial Spearman | -0.180 | -0.002 | -0.217* | -0.049 | -0.265* |
| Predictive $\Delta R^2$ | -0.000 | -0.003 | 0.012 | 0.003 | 0.004 |

*Table 10.* Post fine-tuning (Open Web Math) in-domain loss on (Pile-10K) correlation analysis summary. Values marked with $*$ are significant with $p < 0.05$.

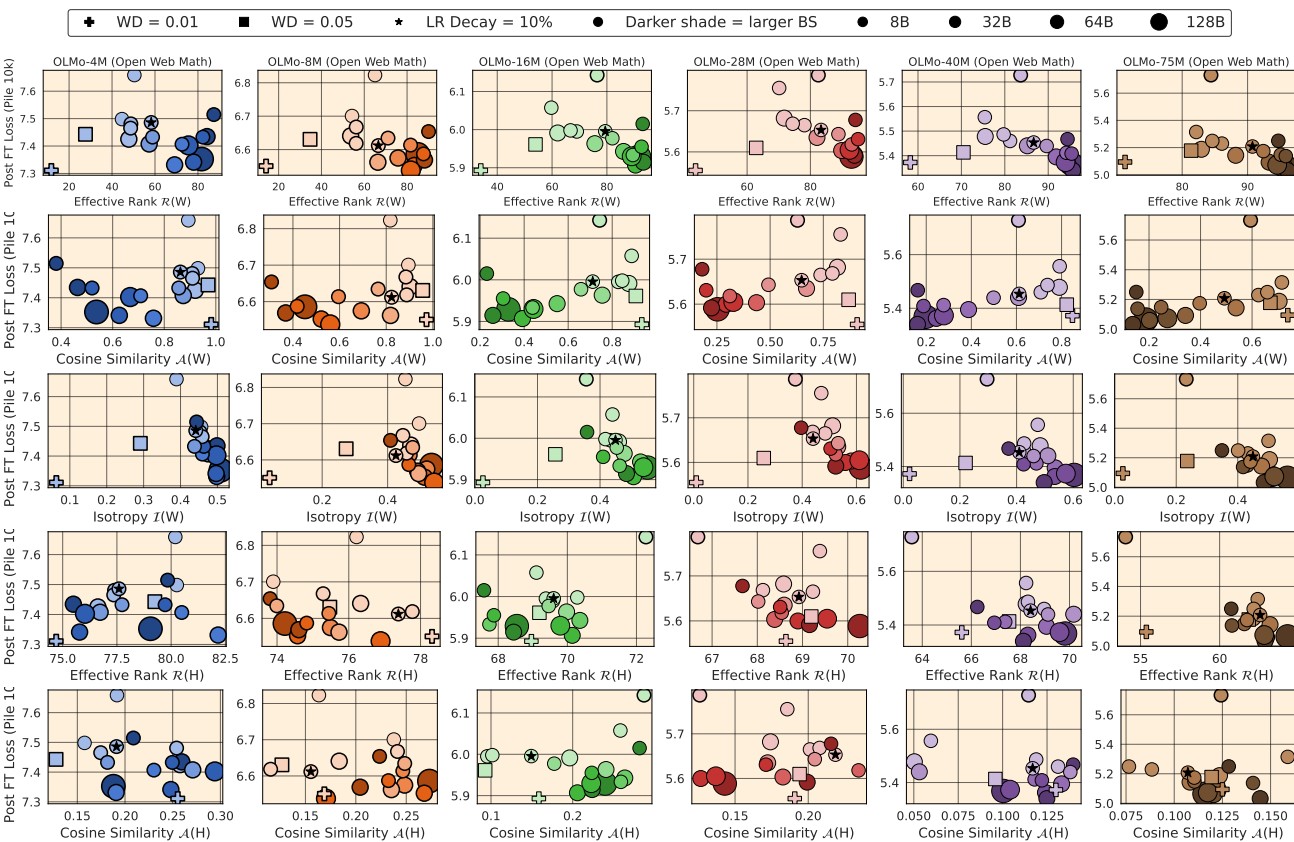

*Figure 27.* Extended forgetting evaluation (OpenWebMath) in relation to geometric metrics of the unembedding matrix and last-layer final-token representation.

## A.4. Extended Discussion: IsoScore

Most geometric metrics that capture the spread of the weights / representations are easily fooled by trivial data transformations. IsoScore (Rudman et al., 2022) fixes this by directly measuring how uniformly variance is distributed across dimensions, giving a reliable $0 - 1$ score where 1 is perfectly even. Specifically, let $\lambda_1 \geq \lambda_2 \geq \cdots \geq \lambda_n \geq 0$ be the eigenvalues of $\mathbf{W}$ that represent the variance along each principal direction. These eigenvalues are then normalized as $\hat{\lambda}_i = \lambda_i / \sum_j \lambda_j$, turning them into a probability distribution over dimensions. A perfectly isotropic distribution would have all $\hat{\lambda}_i = 1/n$, i.e., uniform. To measure how far the actual distribution is from this ideal, IsoScore computes the KL divergence from uniformity,

$$D_{\mathrm{KL}}(\hat{\lambda} \| u) = \sum_{i=1}^{n} \hat{\lambda}_i \log(n \cdot \hat{\lambda}_i),$$

which equals 0 under perfect isotropy and reaches its maximum of $\log n$ when all variance collapses to a single dimension. Finally, this divergence is normalized by $\log n$ and subtracted from 1 to produce a score in $[0, 1]$:

$$IS(W) = 1 - \frac{D_{\mathrm{KL}}(\hat{\lambda} \| u)}{\log n}.$$

A score of $IS(W) = 1$ indicates a perfectly isotropic distribution (a ball), a score of $IS(W) = 0$ indicates maximal anisotropy (a line). The spearman correlation analysis in Table 11 indicates that similar to other geometric metrics, IsoScore of the unembedding matrix is not potent enough

| Metric | Effective Rank | Cosine Similarity | Isotropy | IsoScore |
|---|---|---|---|---|
| Raw Spearman | -0.699 | 0.601 | -0.241 | -0.276 |
| Partial Spearman | -0.261 | -0.100 | -0.346 | 0.018 |

*Table 11.* ID Loss (Pile 10K) correlation analysis summary.

to predict downstream in-domain performance. Moreover, as shown in Figures 28, 29, 30, and 31 it is also heavily associated with hyperparameter trends rather than performance variations.

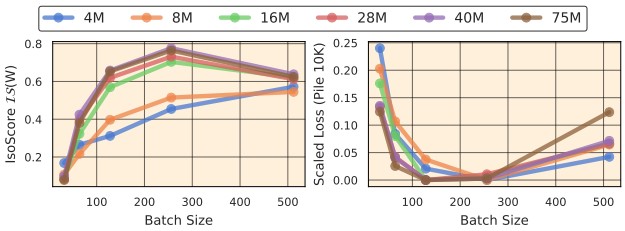

*Figure 28.* Effect of varying batch size on IsoScore.

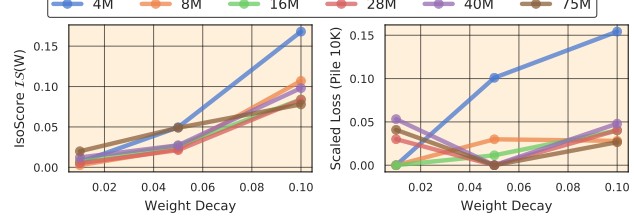

*Figure 29.* Effect of varying weight decay on IsoScore.

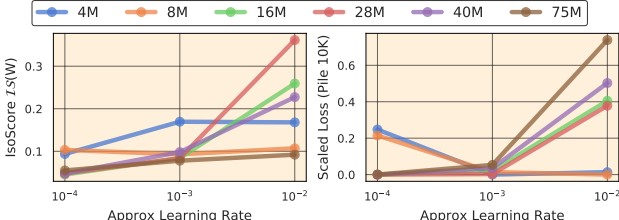

*Figure 30.* Effect of varying learning rate on IsoScore.

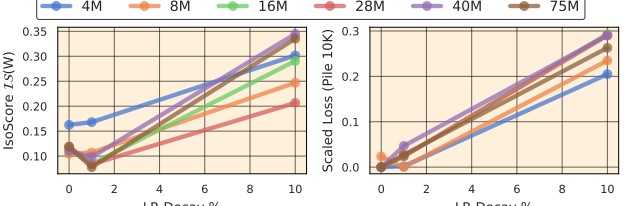

*Figure 31.* Effect of varying LR decay decay on IsoScore.

## A.5. Extended Discussion: Effect of Model Architecture

Across a range of architectural and training variations, we find that the geometry of the unembedding matrix remains largely stable even when downstream performance changes. In particular, Post-LN configurations (after MLP and attention) consistently outperform pre-LN variants (Table 12), yet exhibit similar unembedding geometry, indicating that our observations generalize across normalization choices. Likewise, different activation functions (SwiGLU, ReLU, GeLU) lead to modest differences in performance (Table 13) without materially affecting the unembedding geometry. We observe a similar pattern when varying the pre-training corpus between the Pile and C4 (Table 14), where in-domain performance shifts slightly but geometric properties remain consistent.

| Model | Architecture | ID Loss | Effective Rank % | Cosine Similarity | Isotropy | IsoScore |
|-------|--------------|---------|------------------|-------------------|----------|----------|
| OLMo-4M | Post-LN | 6.59 | 49.74 | 0.91 | 0.45 | 0.23 |
| | Pre-LN | 6.68 | 48.65 | 0.91 | 0.46 | 0.17 |
| OLMo-28M | Post-LN | 4.95 | 75.85 | 0.76 | 0.47 | 0.11 |
| | Pre-LN | 4.97 | 74.40 | 0.79 | 0.44 | 0.08 |

*Table 12.* The effect of **LayerNorm position** on unembedding matrix geometry.

| Model | Activation | ID Loss | Effective Rank % | Cosine Similarity | Isotropy | IsoScore |
|-------|-----------|---------|------------------|-------------------|----------|----------|
| OLMo-4M | ReLU | 6.68 | 48.62 | 0.90 | 0.45 | 0.18 |
| | GeLU | 6.67 | 49.13 | 0.91 | 0.44 | 0.2 |
| | SwiGLU | 6.68 | 48.65 | 0.91 | 0.46 | 0.17 |
| OLMo-28M | ReLU | 5.04 | 77.18 | 0.76 | 0.48 | 0.12 |
| | GeLU | 4.97 | 72.85 | 0.8 | 0.42 | 0.07 |
| | SwiGLU | 4.97 | 74.40 | 0.79 | 0.44 | 0.08 |

*Table 13.* The effect of **different activations** on unembedding matrix geometry.

| Model | Corpus | ID Loss | Effective Rank % | Cosine Similarity | Isotropy | IsoScore |
|-------|--------|---------|------------------|-------------------|----------|----------|
| OLMo-4M | C4 | 8.31 | 50.14 | 0.87 | 0.43 | 0.06 |
| | Pile | 6.68 | 48.65 | 0.91 | 0.46 | 0.17 |
| OLMo-28M | C4 | 6.64 | 75.04 | 0.76 | 0.40 | 0.04 |
| | Pile | 4.97 | 74.40 | 0.79 | 0.44 | 0.08 |

*Table 14.* The effect of **pre-training corpora** on unembedding matrix geometry.

In contrast, tying the input and output embeddings has a more pronounced effect on representation structure (Table 15). Tied embeddings exhibit higher effective rank, greater isotropy, higher IsoScore, and reduced cosine similarity compared to untied embeddings. We hypothesize that tying constrains embeddings to simultaneously serve as input features

| Model | Architecture | ID Loss | Effective Rank % | Cosine Similarity | Isotropy | IsoScore |
|-------|--------------|---------|------------------|-------------------|----------|----------|
| OLMo-4M | tied | 6.78 | 56.15 | 0.87 | 0.52 | 0.25 |
| | untied | 6.68 | 48.65 | 0.91 | 0.46 | 0.17 |
| OLMo-28M | tied | 5.00 | 80.95 | 0.69 | 0.49 | 0.14 |
| | untied | 4.97 | 74.40 | 0.79 | 0.44 | 0.08 |
| OLMo-75M | tied | 4.59 | 87.57 | 0.56 | 0.44 | 0.13 |
| | untied | 4.62 | 84.60 | 0.66 | 0.40 | 0.08 |

*Table 15.* The effect of **weight tying** on unembedding matrix geometry.

and output classifiers, encouraging tokens to become more globally distinguishable and semantically informative, resulting in a more evenly dispersed geometry (Bertolotti & Cazzola, 2024). While OLMo-4M benefits from untied embeddings due to limited capacity (Chung et al., 2021), larger models benefit from tied embeddings, with the 75M model achieving both improved geometry and performance under tying. Overall, despite these differences, the effective rank remains within a comparable range across tied and untied settings.

## B. Practical Takeaways for ML Practitioners

Our systematic investigation of 108 OLmo language models trained under varying hyperparameters reveals that the relationship between model geometry and downstream performance is more complex than previously thought. Hence, we recommend the following practical takeaways for ML Practitioners on how to better assess the model geometry and when it is useful:

1. **Do not use effective rank alone to diagnose model problems.** If one observes a low effective rank during training, this doesn't necessarily indicate the model is failing; it may simply reflect your hyperparameter choices (especially weight decay and batch size). Thus, always measure downstream performance metrics first.

2. **Hyperparameter choices matter more than geometry.** Rather than directly optimizing for geometric properties (e.g., adding explicit rank regularization), focus on tuning standard hyperparameters like batch size and weight decay, which naturally influence geometry and have more predictable effects on performance.

3. **Avoid geometry-based early stopping.** Halting training solely because the effective rank is decreasing is not a good option, as we observe that some of our best-performing models had relatively low effective rank.

4. **Small model saturation may be avoidable.** Unlike Pythia models, our OLMo models do not suffer from late-stage performance degradation even with massive over-training. Thus, architectural choices and data diversity matter a lot, and it's better to optimize on those fronts rather than focusing solely on geometric metrics.

5. **Distinguish correlation from causation.** Our results suggest effective rank is a *symptom* of training dynamics rather than a *cause* of performance differences. Future work should use interventional studies (e.g., directly manipulating rank while holding other factors constant) rather than purely correlational analyses.

## C. Limitations

While our study provides a systematic analysis of unembedding and representation geometry, it has its limitations. First, we focus exclusively on the unembedding matrix and last-token final-layer representations; intermediate layers may exhibit different geometric properties as evidenced by some prior work (Valeriani et al., 2023; Cheng et al., 2025; Skean et al., 2025). Second, our experiments are restricted to relatively small models. Although common in mechanistic studies of Transformers (Wortsman et al., 2024; Kumar et al., 2025; Magnusson et al., 2025; Gadre et al., 2025; Springer et al., 2025), it remains unclear whether these findings generalize to larger models. Third, our fine-tuning experiments hold most hyperparameters fixed, leaving the effects of hyperparameter variation on geometry, performance, and catastrophic forgetting unexplored. Fourth, because directly controlling the model or the representation geometry is challenging, our results are observational rather than fully causal.

Finally, given that (1) we do not train models with extremely large token budgets combined with very small batch sizes, and (2) larger models have higher critical batch sizes, our setup likely places small models beyond their optimal regime more frequently than large models. As a result, small models are more often trained in the over-batched regime, where effective rank remains high despite degraded performance. In contrast, larger models are typically closer to their optimal batch size, where high rank aligns with strong performance. This leads to a weakened observed rank vs performance correlation for smaller models. The partial Spearman correlation scores (which control for all the other hyperparameters) in Table 16 reveal the

| Model | Raw Spearman | Partial Spearman |
|-------|--------------|------------------|
| 4M    | -0.659*      | 0.024            |
| 8M    | -0.695*      | -0.046           |
| 16M   | -0.657*      | -0.244           |
| 28M   | -0.610*      | -0.225           |
| 40M   | -0.707*      | -0.256           |
| 75M   | -0.620*      | -0.483*          |

*Table 16.* Correlations analysis between effective rank and performance across model sizes. Values marked with $*$ are significant with $p < 0.05$.

expected trend: smaller models exhibit weaker correlations than the larger model. However, it is important to note that, except for the OLMo-75M model, these correlations are not statistically significant. Additionally, with only 18 observations per model size and 4–5 control variables, the results should be interpreted cautiously, and strong conclusions cannot be drawn.

