# OpenReview forum: "Disentangling Geometry, Performance, and Training in Language Models"
_ICML.cc/2026/Conference — ICML 2026 spotlight_

### Official Review · Reviewer_Fijr · 2026-03-10

**Soundness:** 3
**Presentation:** 4
**Significance:** 4
**Originality:** 3
**Overall Recommendation:** 5
**Confidence:** 4

**Summary:**

The paper conducts a thorough empirical study on the unembedding matrix’s effective rank and downstream task performance and successfully argues that its presumed predictive power is largely nonexistent. Previous works mistakenly ended up with this result because they overlooked confounders such as batch size and weight decay. It also dives into other related geometric metrics such as cosine similarity and isotropy, and reaches a compelling conclusion that a model’s geometry primarily reflects training choices rather than performance.

**Compliance With Llm Reviewing Policy:**

Affirmed.

**Final Justification:**

Solid contribution

**Key Questions For Authors:**

1. Would accounting for the correlation between model sizes and batch sizes change any of the conclusions?

**Limitations:**

The paper discussed potential negative societal impact. The only technical limitation I noticed is that the correlation between model sizes and batch sizes seems to be under-discussed.

**Strengths And Weaknesses:**

## Soundness

**Strength**

* The paper conducted a thorough empirical study of 108 OLMo-style language models trained under controlled variation, with various model sizes, training set sizes, batch sizes, and weight decays.
* The paper meticulously examines the correlations between numerous metrics including effective dimension, loss, quantization, number of tokens, downstream task loss, etc.
* After concluding that effective ranks do not possess the believed prediction power, the paper looks into confounders such as batch sizes and weight decays, and shows they have better prediction power on downstream task performance.
* The paper extends the approaches to other geometric metrics such as cosine similarity and isotropy, and finally reaches a compelling conclusion that a model’s geometry primarily reflects training choices rather than performance.

**Weakness**

* In Figure 2, there appears to be a correlation between model sizes and batch sizes. It might be determined by scaling law, but the paper does not seem to discuss this correlation and whether it may change the shape of the correlation in Figure 2.

## Presentation
The paper is well written and organizes many sub-research questions in a well-structured way.

## Significance
The paper presents a compelling conclusion that model’s geometry primarily reflects training choices rather than performance.

## Originality
The paper dispelled a misconception that effective rank determines downstream task performance.

---

> ### Author Rebuttal · Authors · 2026-03-31
>
> We thank the reviewer Fijr for their appreciation of our work and thoughtful feedback! Below we address the mentioned weakness / question.
>
> **W1 Correlation between model size and batch size**
>
> 1. This is a very interesting observation by the reviewer which is worth exploring. It is important to note that Figure 2 in the main paper does not represent models trained on larger token budget + smaller batch sizes (large + light colored circles). We do not train such models as they require a high number of optimization steps (eg: training 128B tokens with batch size 32 needs ~2M steps to finish training), necessitating heavy GPU requirements. Thus, for large token budgets, we chose batch sizes such that the optimization steps do not exceed 150k steps. Hence, the hyperparameter ablation for RQ2 is carried out on an 8B token budget. Table 5 in the [link](https://anonymous.4open.science/r/ICML-2026-Rebuttal-5621/_ICML_2026__rebuttal_r1.pdf) details the choice of hyperparameters in our suite.
> 2. Moreover, we use the model-size specific learning rates prescribed by [Porian et al. (2024)](https://arxiv.org/abs/2406.19146)’s scaling laws. But critically, the optimal batch size also follows scaling law prescriptions with larger models having larger optimal batch sizes. Thus, in the main paper Figure 2, the distribution of where each model size sits relative to its critical batch size is systematically different. Small models have more points in the over-batched regime (high rank, degraded performance), while large models have more points near the optimal batch size (high rank, good performance). This would make the rank-performance correlation look weaker for small models and stronger for large models, given our experimental design.
> 3. Below we present the correlation analysis stratified by model size. The partial Spearman correlation scores (which control for all the other hyperparamters) reveal the expected trend: smaller models exhibit weaker correlations than the larger model. However, it is important to note that, except for the OLMo-75M model, these correlations are not statistically significant. Additionally, with only 18 observations per model size and 4–5 control variables, the results should be interpreted cautiously, and strong conclusions cannot be drawn.
>
> | Model  | Raw Spearman | Partial Spearman   |
> | :--:|:--:|:--:|
> | 4M | -0.659* | 0.024 |
> | 8M | -0.695* | -0.046  |
> | 16M | -0.657* | -0.244 |
> | 28M | -0.610*| -0.225 |
> | 40M | -0.707* | -0.256  |
> | 75M | -0.620*  | -0.483*  |
> *Significance levels:* (*p < 0.05)
>
> 4. Thus, we hypothesize that a broader sweep of hyperparamters across the larger token budget and smaller batch size would reduce the correlation on the larger models in our suite.

---

> > ### Author Rebuttal · Reviewer_Fijr · 2026-04-04
> >
> > Thanks to the authors for the rebuttal. My concerns have been sufficiently addressed, and I will maintain my score. The paper remains a solid contribution.

---

> > > ### Author Response · Authors · 2026-04-05
> > >
> > > Thank you very much!
> > >
> > > We are delighted that our rebuttal has addressed your concerns. We will definitely include the rebuttal content in the updated manuscript.
> > >
> > > We sincerely appreciate your time and effort!

---

### Official Review · Reviewer_DT3f · 2026-03-12

**Soundness:** 4
**Presentation:** 3
**Significance:** 2
**Originality:** 2
**Overall Recommendation:** 4
**Confidence:** 3

**Summary:**

This paper studies whether the geometry of the unembedding matrix, which has been widely used in interpretability research, actually provides meaningful insight into downstream model quality. To answer this, the paper systematically studies the relationship between model performance and unembedding geometry across a controlled suite of models, aiming to clarify whether commonly used geometric properties are genuine indicators of capability or merely reflections of training dynamics and hyperparameter choices.

**Compliance With Llm Reviewing Policy:**

Affirmed.

**Final Justification:**

The authors’ response is greatly appreciated and addresses most of my concerns. However, some concerns regarding the evaluation limitations still remain. I encourage the authors to include the extended discussion in the final version.

**Key Questions For Authors:**

1) The paper argues that existing geometric metrics are not reliable predictors of downstream performance. If so, what underlying latent factor are these metrics actually capturing, and can the authors quantify this more directly? At present, it is not yet clear that hyperparameters such as batch size, weight decay, and learning rate are the true driving factors, especially without stronger evidence at larger scales.

2) Why does higher effective rank sometimes correlate with better performance? A more detailed explanation of when and why this relationship appears would strengthen the paper.

3) Since this is primarily an empirical study, can the authors include similar experiments on larger models and a broader range of downstream tasks to better assess the generality of the conclusions?

**Limitations:**

yes

**Strengths And Weaknesses:**

Strengths

1) This paper usefully challenges an overly simplistic rank–performance narrative and shows that commonly used geometric proxies can be substantially confounded by training choices.

2) The paper is clearly written, and the presentation is easy to follow.

Weaknesses

1) A main limitation is that the study is conducted primarily on relatively small-scale models. Even within these experiments, the results suggest that the effect of hyperparameters may weaken as model size increases; for example, Figure 8 indicates that the impact of batch size appears to diminish at larger scales.

2) I do not see a clear actionable takeaway from the paper. While the work questions existing geometric proxies, it is less clear what practitioners should use instead.

3) Recent works have already studied representation geometry, but the motivation here does not feel fully developed beyond the claim that representation geometry varies less than unembedding geometry.

4) The work lacks a strong theoretical or mechanistic justification for why unembedding geometry should be expected to correlate with downstream task performance in the first place. Without such motivation, the empirical study feels somewhat exploratory.

5) The evaluation benchmarks also seem somewhat limited, which makes it harder to assess how broadly the conclusions generalize.

---

> ### Author Rebuttal · Authors · 2026-03-31
>
> We thank the reviewer DT3f for their positive comments & thoughtful feedback! Below we address each of the weaknesses.
>
> **W1 Model Sizes**
>
> Prior work [1,2] shows that low effective rank primarily arises in smaller language models where V  >> d, so we focus on training and evaluating small LMs, which is also standard in LLM interpretability studies [3,4].
> We agree that as per Figure 8, the effective rank changes little for larger models when changing the batch size. A key reason is that larger models inherently have a richer singular value spectrum due to high d. As a result, achieving a comparable level of rank collapse to that of a smaller model requires suppressing many more dimensions. For e.g., a 50% collapse corresponds to removing 32 dimensions when d=64, but 384 dimensions when d=768. This makes large models less sensitive to hyperparameter-induced changes in effective rank.
>
>
> **W2 Takeaway**
>
> The main takeaway is that effective rank is not a reliable standalone signal for model quality. It mostly reflects hyperparameter choices rather than causing performance differences. Thus, In practice, one should focus on tuning standard hyperparameters (e.g., batch size, weight decay) and evaluating downstream metrics, rather than optimizing geometric properties directly. We elaborate on this in Appendix B.
>
> **W3 Representation vs Unembedding Geometry**
>
> We examine representation geometry (H) largely because prior work [1,2] uses it alongside unembedding geometry (W). Our findings go beyond showing that the geometry of H varies less than W: we show that the two capture different properties. The effective rank of H reflects variation in final-layer representations across inputs, whereas W is directly shaped by gradients and encodes prediction errors. Since H arises from layered transformations and W from the output objective, there is no strong reason to expect their effective ranks to correlate. Thus, we conclude that they should not be analyzed interchangeably.
>
> **W4 Theoretical justification**
>
> The unembedding matrix W maps hidden states to logits. So low rank W can limit the diversity of output distributions, creating a capacity bottleneck. This has been studied extensively both theoretically & experimentally [1, 5, 6]. Instead, we actually argue against using effective rank as a predictor of performance as it measures how information is distributed across the singular spectrum of W, not how well directions are utilized or how much task-relevant information is encoded. Our empirical results therefore motivate a more rigorous theoretical analysis to characterize when a high effective rank in W improves performance, and when it does not.
>
> **W5 Limited Evaluation**
>
> While broader evaluation would be valuable, our setup already probes multiple aspects of LLM behavior in a unified way. We only focus on tasks that use the unembedding matrix; classification-style tasks are excluded since they replace it with a task-specific head. Additionally, complex tasks (e.g., reasoning, coding) are beyond the scope given our model scale and lack of post-training.
>
> **Q1**
> As mentioned in W3 geometric metrics capture distributional spread of the singular value spectrum, not necessarily task-relevant information. Tthis spectrum is known to be shaped by optimization dynamics. For example, large batch sizes average gradients over many tokens, reducing frequency biases and keeping W spread across singular directions. But excessive averaging lowers the signal-to-noise ratio, preventing fine-grained learning. Thus, W can becomes high-rank but informationally diffuse, similar to randomly initialized models. Likewise, strong weight decay prevents any specific direction from dominating, thus, increasing rank but reducing performance. Why do these hyperparameters affect less at larger sizes is elaborated in W1.
>
> **Q2**
> While the observation is true, we do not attribute high effective rank to cause good performance. As mentioned in the paper, we find well-performing models result from favorable hyperparameters, with high effective rank emerging as a byproduct. Moreover, rank suitability also depends on the task: low rank might suffice for classification or clustering, where outputs are needed to concentrate in a few directions; whereas high rank may be suitable for open-ended tasks to support diverse outputs & capture the variation in the model’s predictions.
>
> **Q3**
> See response to W1 and W5
>
>
> **References**
> 1. Why do small language models underperform? Studying Language Model Saturation via the Softmax Bottleneck, Godey et al. 2024
> 2. Anisotropy is Not Inherent to Transformers, Machina & Mercer 2024
> 3. Small-scale proxies for large-scale Transformer training instabilities , Wortsman et al., 2024
> 4. Overtrained Language Models Are Harder to Fine-Tune  Springer et al., 2025
> 5. Breaking the Softmax Bottleneck: A High-Rank RNN Language Model Yang et al., 2018
> 6. Closing the Curious Case of Neural Text Degeneration Finlayson et al., 2023

---

> > ### Author Rebuttal · Reviewer_DT3f · 2026-04-04
> >
> > The authors’ response is greatly appreciated and addresses most of my concerns. However, some concerns regarding the evaluation limitations still remain. I will raise my score to 4. I encourage the authors to include the extended discussion in the final version.

---

> > > ### Author Response · Authors · 2026-04-05
> > >
> > > Thank you very much!
> > >
> > > We are delighted that our rebuttal has addressed your concerns. We will definitely include the rebuttal content in the updated manuscript.
> > >
> > > We sincerely appreciate your time and effort!

---

### Official Review · Reviewer_Ci6i · 2026-03-12

**Soundness:** 3
**Presentation:** 3
**Significance:** 3
**Originality:** 3
**Overall Recommendation:** 5
**Confidence:** 4

**Summary:**

Past work has suggested low effective rank of the unembedding matrix in LLMs to drive high performance. This study asks whether the relationship is correlational or causal, and attempts to isolate the reasons for high or low effective rank. The authors find that effective rank correlates to, but does not causally drive, LLM performance. They show that high or low effective rank results from a number of choices during pre-training (e.g., batch size). Finally, effective rank highly correlates to other geometric properties of interest, for instance isotropy. This suggests effective rank as a diagnostic tool, but not a causal factor, in evaluating LLM performance.

**Compliance With Llm Reviewing Policy:**

Affirmed.

**Final Justification:**

My concerns have been addressed and I am keeping my high score.

**Key Questions For Authors:**

N/A

**Limitations:**

yes

**Strengths And Weaknesses:**

## Strengths

This is an extremely thorough analysis of how the effective rank of the unembedding matrix relates to performance (pretraining, OOD loss, finetuning, forgetting, quantization). Experiments are very large-scale (108 pre-trained language models). The takeaway that effective rank can be helpful, but is ultimately inadequate to diagnose model performance is helpful to practitioners, and resolves past debates in the literature to some extent. The takeaway that effective rank is a reflection of hyperparameter choices is also a useful takeaway, which suggests one should directly consider hyperparameters instead of centering geometry.

## Weaknesses

The weaknesses of the work are largely addressed by the authors in Appendix C, Limitations.

Tables 2 and 3 are missing p-values for the Spearman correlations.

---

> ### Author Rebuttal · Authors · 2026-03-31
>
> Thanks to the reviewer Ci6i for their thoughtful comments and positive feedback! Here are the requested p-values:
>
> For table 2 (In distribution performance evaluation on Pile 10k)
> | Metric                         | Effective Rank R(W) | Cosine Similarity A(W) | Isotropy I(W) | IsoScore IS(W) | Effective Rank R(H) | Cosine Similarity A(H) | IsoScore IS(H) |
> |----------------------------------|--------------------|------------------------|---------------|----------------|---------------------|------------------------|----------------|
> | Raw Spearman ρ                   | -0.699*          | 0.601*              | -0.241*       | -0.276*       | 0.773*            | 0.388*              | 0.700*       |
> | Residual Spearman (linear)       | -0.209*            | 0.233*                | -0.155        | -0.220*        | -0.008              | -0.174                | 0.024          |
> | Residual Spearman (Hoffmann)     | -0.724*          | 0.635*              | -0.333*     | -0.332*      | 0.729*            | 0.337*              | 0.661*       |
> | Partial Spearman                 | -0.232*            | -0.102                | -0.307*      | 0.024          | -0.052              | -0.102                | 0.442*       |
> | Predictive ΔR²                  | -0.004             | -0.006                | 0.035         | -0.003         | 0.004               | -0.003                | 0.001          |
>
>
> For table 3 (out-of-distribution performance evaluation on Dolma 100 code)
> | Metric                         | Effective Rank R(W) | Cosine Similarity A(W) | Isotropy I(W) | IsoScore IS(W) | Effective Rank R(H) | Cosine Similarity A(H) | IsoScore IS(H) |
> |----------------------------------|------------------------------------------|--------------------------------------------|-----------------------------------|------------------------------------|------------------------------------------|--------------------------------------------|------------------------------------|
> | Raw Spearman $\rho$               | -0.672*                                | 0.577*                                   | -0.182                            | -0.267*                            | 0.845*                                | 0.127                                      | 0.770*                           |
> | Residual Spearman (linear)        | -0.049                                   | 0.073                                      | -0.084                            | -0.061                              | 0.009                                   | -0.106                                     | 0.053                              |
> | Residual Spearman (Hoffmann)      | -0.692*                                | 0.606*                                   | -0.261*                           | -0.316*                            | 0.809*                                | 0.098                                      | 0.734*                           |
> | Partial Spearman                  | -0.157                                   | -0.201                                     | -0.243*                           | 0.144                               | 0.103                                   | -0.063                                     | 0.607*                           |
> | Predictive $\Delta$R$^2$          | -0.015                                   | 0.005                                      | 0.019                             | -0.008                              | -0.004                                  | -0.010                                     | 0.043                              |
>
> *Significance levels:* (*p < 0.05)
>
> While raw Spearman correlation is almost always significant for Effective rank when comparing ID or OOD performance, that is not always the case for partial spearman correlation. This further weakens the influence of effective rank in being a reliable predictor of downstream performance.
> We are happy to address any questions you may have!

---

> > ### Author Rebuttal · Reviewer_Ci6i · 2026-04-01
> >
> > This is great! Thanks.

---

> > > ### Author Response · Authors · 2026-04-05
> > >
> > > Thank you very much!
> > >
> > > We are delighted that our rebuttal has addressed your concerns. We will definitely include the rebuttal content in the updated manuscript.
> > >
> > > We sincerely appreciate your time and effort!

---

### Official Review · Reviewer_Zk4q · 2026-03-13

**Soundness:** 3
**Presentation:** 3
**Significance:** 3
**Originality:** 3
**Overall Recommendation:** 5
**Confidence:** 4

**Summary:**

This paper presents an empirical study of the relationship between unembedding matrix geometry and language model performance. The authors train OLMo-style models under controlled hyperparameter variation and evaluate five downstream axes including in-distribution loss, OOD generalization, fine-tuning, knowledge retention, and post-training quantization. The central argument is that effective rank tends to correlate with performance but is neither necessary nor sufficient to predict it, and that geometric metrics primarily reflect training choices rather than model quality. The paper also revisits the saturation phenomenon, arguing that low effective rank co-occurs with rather than causes late-stage performance degradation.

**Compliance With Llm Reviewing Policy:**

Affirmed.

**Final Justification:**

Reviewers have fully addressed my concerns. I now believe the paper should be accepted.

**Key Questions For Authors:**

- The paper finds that weight decay is the dominant driver of effective rank across model sizes. Does this mean effective rank could be repurposed as a cheap diagnostic signal for whether weight decay is in the appropriate regime for a given model size, even if it cannot predict absolute performance?

**Limitations:**

yes.

**Strengths And Weaknesses:**

__Strengths__

- The scale and systematic design of the study is the paper's clearest contribution. Training 108 models under controlled ablations across batch size, weight decay, learning rate, and token budget gives the authors a nice setup  to disentangle confounded effects that prior single-model or single-family studies cannot address.

- The reframing of the saturation finding is valuable. Demonstrating that OLMo models can exhibit severely low effective rank without saturation, and that saturation-like behavior can occur without low effective rank, is a meaningful extension of prior causal claims.

- The multi-task evaluation scope is broader than most prior geometry papers, which tend to focus on pretraining loss alone. Including quantization robustness is particularly timely and practically useful.

- The finding that geometry captures hyperparameter choices more than performance is clearly articulated and well-supported empirically. This is an important epistemic caution for the interpretability community.

__Weaknesses__

- The isotropy metric used throughout is the partition-function-based score from Arora et al., which the authors call ``isotropy'' but which has been shown to be an unreliable measure of true isotropy [1]. This is a significant methodological concern. The IsoScore literature demonstrates that partition-based scores conflate anisotropy with other geometric properties, meaning the isotropy-related conclusions in Section 5 should be treated with caution. The authors should have used IsoScore [1] and the absence of any engagement with this line of criticism weakens the geometric analysis section considerably.

- The authors do mention that Mickus et al. [2] show that anisotropy is not detrimental for clustering-based tasks and is often a byproduct of meaningful representational structure rather than a failure mode, however a deeper comparison with their claims is needed. Further, Rudman et al. 2024 shows that anisotropy can be beneficial for model performance. [3] The paper's treatment of anisotropy as broadly problematic does not adequately engage with this result.

- All experiments are restricted to OLMo-style architectures trained on the Pile. It remains unclear whether the findings generalize to models with tied embeddings, different tokenizer vocabularies, or architectures that do not use a separate unembedding matrix.

- [1] IsoScore: Measuring the Uniformity of Embedding Space Utilization. Rudman et al. 2022
- [2] Isotropy, Clusters, and Classifiers. Mickus et al. 2024.
- [3] Stable Anisotropic Regularization. Rudman et al. 2024.

---

> ### Author Rebuttal · Authors · 2026-03-31
>
> We thank the reviewer Zk4q for their positive comments and thoughtful feedback! Below we address each of the weaknesses.
>
> **W1 Isoscore Metric**
>
> Thank you for recommending the IsoScore metric and related work (Rudman et al., 2024); we will cite these. Our findings on IsoScore are as follows:
>
> 1. The correlation analysis with ID loss is shown below. The low partial Spearman scores indicating weak predictive power of IsoScore for ID performance.
> |Metric|Effective Rank|Cosine Similarity|Isotropy|IsoScore|
> |:--:|:--:|:--:|:--:|:--:|
> | Raw Spearman |-0.699|0.601|-0.241|-0.276|
> | Partial Spearman|-0.261|-0.100|-0.346|0.018|
> 2. The figures/tables ([link](https://anonymous.4open.science/r/ICML-2026-Rebuttal-5621/_ICML_2026__rebuttal_r1.pdf)) shed more light on how IsoScore fares with other metrics, the effect of hyperparameters on it, & its evolution with model saturation.
> In summary, IsoScore is more expressive than partition-function based Isotropy and correlates better with effective rank & cosine similarity. However, like other geometric metrics, it is also heavily associated with hyperparameter trends and does not reliably explain performance changes.
>
> **W2: Comparison with Mickus et al., 2024**
>
> While Mickus et al. (2024) and Rudman et al. (2024) show anisotropy can help in some tasks, their scope and evaluation differ from ours.
> 1. We focus on decoder-only models, tasks involving next-token prediction, and study how the unembedding matrix geometry relates to performance. In contrast, they study encoder-only models (ALBERT, DistilBERT, S-BERT) on classification tasks (SST-2, QNLI, Squad, MRPC,etc) which use use linear classification head instead of an unembedding matrix. Thus, their findings about the geometry of encoder-only representations cannot be directly compared to our decoder-only findings.
> 2. Additionally, we do not treat anisotropy as problematic; instead, we question its reliability in predicting performance. Unlike these works, we do not posit that the geometry (effective rank / cosine similarity / isotropy) of  the unembedding matrix or last token’s final-layer representation to be inherently predictive of performance. Rather, we show geometry changes are associated with hyperparameter choices, which in turn can affect performance.
>
> **W3: Other Model Architectures**
>
> We would like to clarify that we call our models 'OLMO-style’ only because we use the OLMo code base as our setup. Thus, our model architecture can be easily tweaked with making changes to their code base. Here is an analysis of how effective rank changes with different architectural changes ([figures/tables link](https://anonymous.4open.science/r/ICML-2026-Rebuttal-5621/_ICML_2026__rebuttal_r1.pdf)).
> 1. LayerNorm (pre- vs post-LN): Post-LN (after MLP and attention) consistently outperforms pre-LN (Table 1), yet unembedding geometry remains stable, suggesting results generalize across architectures.
> 2. Activations (SwiGLU, ReLU, GeLU): The performance varies slightly among the different activations (Table 2). Nevertheless, the geometry of the embedding matrix remains stable.
> 3. Pre-training corpus (Pile vs C4): Similar to previous results, ID performance varies slightly but the unembedding geometry remains stable (Table 3). These three analyses support the generalizability of our findings about the unembedding matrix geometry.
> 4. Embeddings (tied vs untied):
> From Table 4, we can see that tying the embeddings exhibit higher effective rank, isotropy, IsoScore, and reduced cosine similarity compared to their untied counterparts. We conjecture that tying forces embeddings to  act as features on the input side and as classifiers on the output side, thus, creating an implicit constraint on the tokens to be globally distinguishable and semantically useful, rendering a more spreading out geometry without favouring any particular direction strongly ([Bertolotti & Cazzola, 2024](https://openreview.net/forum?id=yyYMAprcAR)).
> OLMo 4M benefit from untied embeddings due to capacity limits ([Chung et al., 2020](https://arxiv.org/abs/2010.12821)), but as size grows, tied embeddings improve both geometry and performance, with 75M tied models outperforming untied.
> In summary, we conclude that the effective rank of tied and untied models remains in a similar range.
>
> **Q: Effective rank for weight decay**
>
> We agree that  effective rank is sensitive enough to weight decay to be useful diagnostically. If one observes an unusually low effective rank, then increasing the regularization strength would help inhibit stronger updates along specific singular directions, thus, increasing the effect rank.
> However, we caution against over-interpreting this signal for two reasons. 1) Weight decay is not the only hyperparameter shaping effective rank; batch size also influences effective rank. 2) While well-performing models tend to exhibit higher effective rank, this correlation is not universal across tasks and training configurations as shown in our paper.

---

> > ### Author Rebuttal · Reviewer_Zk4q · 2026-04-01
> >
> > Thank you for the detailed response. I believe that you have fully addressed my concerns and I will happily raise my score.

---

> > > ### Author Response · Authors · 2026-04-05
> > >
> > > Thank you very much!
> > >
> > > We are delighted that our rebuttal has addressed your concerns. We will definitely include the rebuttal content in the updated manuscript.
> > >
> > > We sincerely appreciate your time and effort!

---

### Decision · Program_Chairs · 2026-04-30

**Decision:**

Accept (spotlight)

**Comment:**

The reviewers were uniformly positive. They highlighted the scale and systematic design of the study and viewed the central takeaway as useful: effective rank of the unembedding matrix depends on training choices such as batch size and weight decay and should not be overinterpreted as standalone proxy for performance.

Initial concerns about choice of isotropy metric, generalization beyond the studied model family, and possible confounding in the experimental setup were addressed these well. All reviewers indicated their concerns were resolved.

I support acceptance. I do not see the paper as overturning a deeply compelling prior belief, but its value is that it takes a claim from recent work and subjects it to a very careful, large-scale empirical test. Personally, I would like to see more papers of this kind that include careful tests of widely-repeated claims, before they become community belief.

I encourage the authors to sharpen two things in the final version. First, the paper would benefit from discussing more directly why its negative result is useful even if not especially surprising: as the authors note in rebuttal, "effective rank describes how mass is distributed across the singular spectrum of W, but not whether those directions are used or how much task-relevant information they encode", so it is unsurprising that a single scalar fails to predict performance. The lesson then seems indeed that more refined geometric characterizations such as alignment or cosine-similarity-based, should matter more than rank-based summaries, and the paper would benefit from more explicit connections to that line of recent work more explicitly. Second, the treatment of embedding geometry H feels secondary to that of W; since downstream behavior depends on both the readout and the representations being read out, a more symmetric treatment, or a clearer justification for centering W, would I think make the paper feel more complete.

Overall, all reviewers and the AC agree this is a careful and useful empirical contribution. I recommend accept.